# INDICVISIONBENCH: BENCHMARKING CULTURAL AND MULTILINGUAL UNDERSTANDING IN VLMS

**Ali Faraz[1], Akash[2], Shaharukh Khan[1], Raja Kolla[1], Akshat Patidar[1], Suranjan Goswami[2], Abhinav Ravi[1], Chandra Khatri[1], Shubham Agarwal[1]**

[1]*Krutrim AI, Bangalore, India*

[2]*OLA Electric, Bangalore, India*

Contact: {ali.faraz, raja.kolla, shubham.agarwal1}@olakrutrim.com

## ABSTRACT

Vision-language models (VLMs) have demonstrated impressive generalization across multimodal tasks, yet most evaluation benchmarks remain Western-centric, leaving open questions about their performance in culturally diverse and multilingual settings. To address this gap, we introduce *IndicVisionBench*, the first large-scale benchmark centered on the Indian subcontinent. Covering English and 10 Indian languages, our benchmark spans 3 multimodal tasks, including Optical Character Recognition (OCR), Multimodal Machine Translation (MMT), and Visual Question Answering (VQA), covering 6 kinds of question types. Our final benchmark consists of a total of 5K images and 37K+ QA pairs across 13 culturally grounded topics. In addition, we release a paired parallel corpus of annotations across 10 Indic languages, creating a unique resource for analyzing cultural and linguistic biases in VLMs. We evaluate a broad spectrum of 8 models, from proprietary closed-source systems to open-weights medium and large-scale models. Our experiments reveal substantial performance gaps, underscoring the limitations of current VLMs in culturally diverse contexts. By centering cultural diversity and multilinguality, IndicVisionBench establishes a reproducible evaluation framework that paves the way for more inclusive multimodal research. Our benchmark is publicly available at `https://huggingface.co/datasets/krutrim-ai-labs/IndicVisionBench`.

## 1 INTRODUCTION

Vision-language models (VLMs) (Bai et al., 2023; Chen et al., 2024; Lu et al., 2024; Wang et al., 2024b; Laurençon et al., 2024; Tong et al., 2024; Xue et al., 2024) have demonstrated strong performance across a variety of multimodal tasks. However, existing benchmarks (Antol et al., 2015; Fu et al., 2023; Goyal et al., 2017) remain heavily Western-centric, limiting our understanding of how these models generalize to culturally diverse and multilingual settings. India, in particular, represents one of the most culturally and linguistically diverse regions globally, with 22 official languages and 28 states plus 8 Union Territories[1] , each with distinct ethnic, visual, and cultural identities. While some recent efforts partially cover this diversity (Romero et al., 2024; Nayak et al., 2024; Vayani et al., 2025), a systematic, large-scale benchmark capturing India-specific cultural concepts across multiple languages is still lacking.

To address this gap, we introduce **IndicVisionBench**, a culturally grounded evaluation benchmark tailored for the Indian subcontinent. To the best of our knowledge, this is the first large-scale benchmark explicitly designed to assess VLMs in the context of Indian culture and languages. We use states as a proxy for cultural groups following prior works (Adilazuarda et al., 2024; Nayak et al., 2024). IndicVisionBench comprises 5K unique images and 37K+ question-answer pairs spanning 13 cultural topics, covering English and 10 medium-to-low resource Indic languages supporting three multimodal tracks: *Visual Question Answering (VQA)*, *Optical Character Recognition (OCR)*, and

---

[1]`https://en.wikipedia.org/wiki/States_and_union_territories_of_India`

*Multimodal Machine Translation (MMT).* Figure 1 illustrates examples reflecting diverse cultural nuances, including monuments, food, and digitized text. Rigorous human verification and correction at every stage of data collection ensure the reliability and cultural fidelity of the benchmark, covering medium-to-low resource languages including Hindi, Bengali, Tamil, Malayalam, Telugu, Marathi, Kannada, Gujarati, Punjabi, and Oriya.

In this study, we evaluate 8 state-of-the-art (SOTA) VLMs on IndicVisionBench and find that performance drops considerably for low-resource languages and culturally specific content. We also observe a clear gap between proprietary and open-source models in their ability to capture linguistic and cultural nuances across multimodal tasks. Analysis across scripts and language groups further highlight the need for better support and representation of underrepresented regions and cultures.

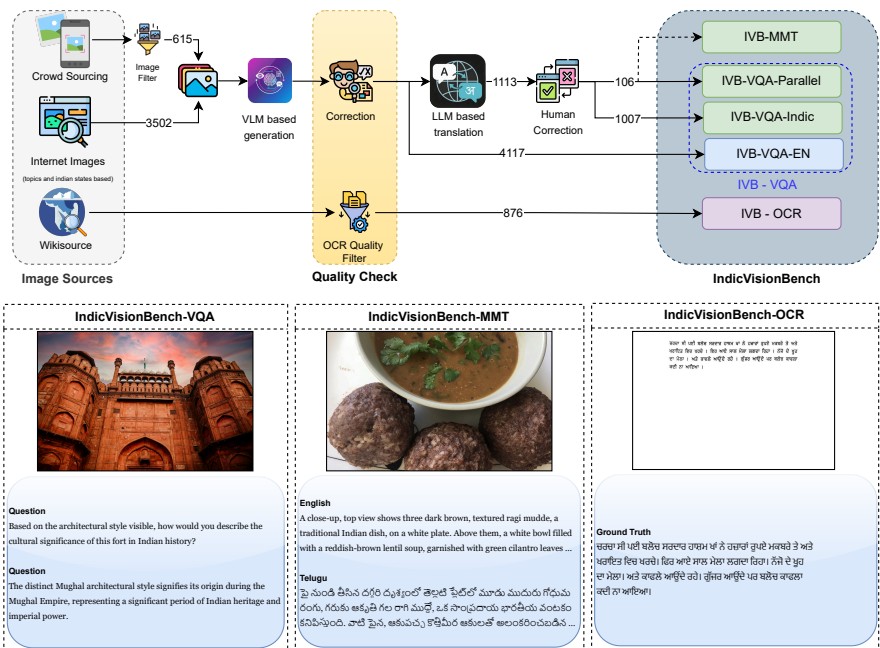

Figure 1: **IndicVisionBench (IVB) pipeline and 3 tracks.** Top panel illustrates our image collection pipeline for 10 Indian languages, showing the number of images at each step, with human quality checks applied throughout. We also present sample outputs for the three tracks: VQA (Visual Question Answering) in English, MMT (Multimodal Machine Translation) in Telugu, and OCR (Optical Character Recognition) in Punjabi. Further details are provided in Section 3.

Our contributions could thus be summarized as follows:

- We propose *IndicVisionBench* as the first large-scale, Indian-centric benchmark for evaluating VLMs on culture-specific understanding, involving OCR, recognition, cultural identification, multi modal translation and semantic understanding involving 5K unique images.

- We conduct a comprehensive evaluation of 8 prominent closed-source as well as open-weight models supporting Indian languages and contrast their performance across all the 3 tracks. We highlight systematic performance gaps that underscore the limitations of current general-purpose VLMs in culturally diverse settings.

- We systematically study the regional-language biases, performance across topics and cross-lingual variation in performance. Our benchmark and relevant code are publicly available for future research in this direction.

## 2 RELATED WORK

**Vision Language Models and Benchmarks.** Cross-attention models (Alayrac et al., 2022; Singh et al., 2022) and later *visual instruction tuning* based auto-regressive models like the LLaVA family (Liu et al., 2023a; 2024), have advanced multimodal learning, where vision encoders (Radford et al., 2021; Zhai et al., 2023; Tschannen et al., 2025) are aligned with large language models. This approach has since influenced a range of VLMs (Lu et al., 2024; Laurençon et al., 2024; Tong et al., 2024; Xue et al., 2024; Team et al., 2024), which follow similar design principles and achieve strong results on translation, captioning, and multi-turn vision language benchmarks (Hudson & Manning, 2019; Fu et al., 2023; Yu et al., 2023). In contrast, multimodal models that handle Indic languages remain relatively underexplored. Most open-source systems provide support only for 2 to 4 medium-resource Indian languages (Maaz et al., 2024; Alam et al., 2025; Yue et al., 2025), with the notable exception of Chitrarth (Khan et al., 2025b), which extends coverage to all ten languages, considered in this work. We include all these models in our benchmark to assess their relative strengths.

**Optical Character Recognition (OCR).** OCR has progressed from early rule-based engines such as Tesseract (Smith, 2007) to modern transformer-based approaches like TrOCR (Li et al., 2021) and docTR (Liao et al., 2023). Recent efforts in document understanding further leverage multimodal architectures, including the DocOwl series (Hu et al., 2024a;b), DocLLM (Wang et al., 2023), and Donut (Kim et al., 2022). These systems are typically evaluated on benchmarks such as RVL-CDIP (Harley et al., 2015), FUNSD (Jaume et al., 2019), and DocVQA (Mathew et al., 2021), to name a few. Despite this progress, existing OCR benchmarks are largely English-centric, offering minimal coverage of Indic scripts and multilingual contexts.

**Multimodal Machine Translation (MMT).** Recently, Multimodal Machine Translation (MMT) (Calixto & Liu, 2017; Elliott & Kádár, 2017; Delbrouck & Dupont, 2017; Yao & Wan, 2020) has gained traction, where the translation leverages auxiliary modalities (e.g., images). Prior works have largely centered on English-European language pairs (Elliott et al., 2016; Specia et al., 2016), with a subset of medium-resource Indian languages (particularly Hindi, Bengali, Malayalam) also explored in the shared task series (Nakazawa et al., 2019; 2020; 2021; 2022; 2023) based on Visual Genome images (Krishna et al., 2017). We support a similar task based on a diverse set of cultural images avoiding potential data contamination issues (Balloccu et al., 2024).

**Cultural VQA.** Several benchmarks have begun probing cultural and multilingual reasoning in VLMs. GD-VCR (Yin et al., 2021) and Henna (Alwajih et al., 2024) emphasize culturally specific content but are largely limited to English or Arabic, while WorldCuisines (Winata et al., 2025) focuses on food and cuisines. Multilingual benchmarks (Liu et al., 2023b; Zhang et al., 2023; Sun et al., 2024; Das et al., 2024; Wang et al., 2024a; Fu et al., 2024a) expand language coverage but often lack cultural and task diversity. Datasets like MaRVL (Liu et al., 2021) and xGQA (Pfeiffer et al., 2021) broaden multilingual reasoning but do not incorporate Indic cultural grounding. Closest to our work are CVQA (Romero et al., 2024), CulturalVQA (Nayak et al., 2024), and ALM-Bench (Vayani et al., 2025), which partially touch on India-specific contexts, yet none offers a unified framework capturing both Indic cultural diversity and multilingual multimodal evaluation.

## 3 BENCHMARK CREATION

Figure 1 illustrates our curation pipeline across all tracks; additional details are provided below.

### 3.1 INDICVISIONBENCH-VQA

We constructed the VQA split using two approaches: (i) controlled crowd-sourcing and (ii) large-scale web crawling. In the first phase, we recruited volunteers (including authors) who contributed images captured on their personal devices along with corresponding annotations. These images were further reviewed to determine whether they were culturally specific to India and, if so, mapped to one of 13 predefined topics and to the relevant State/Union Territory (UT). Irrelevant images were discarded, resulting in 615 valid samples. As several categories and regions were underrepresented, we expanded coverage in the second phase, where cultural experts systematically collected Creative

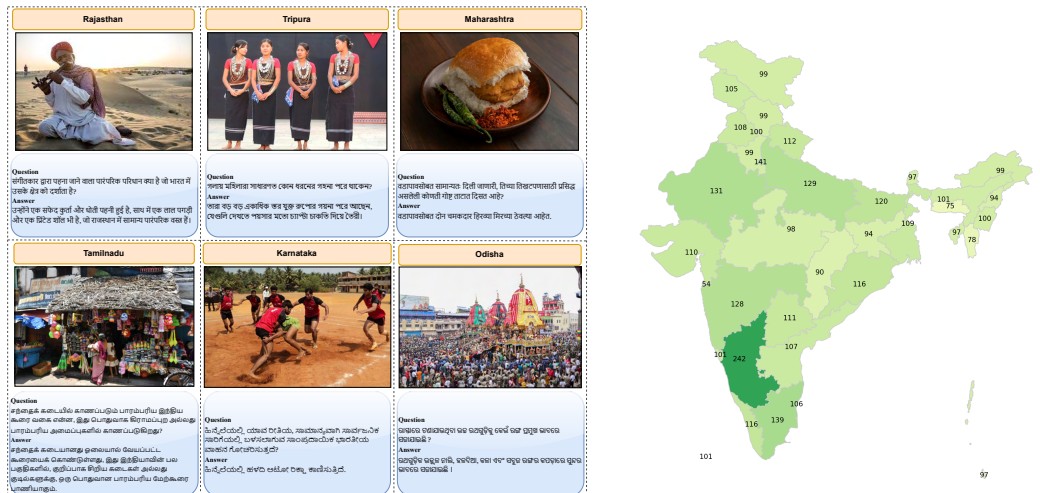

Figure 2: **Examples from IndicVisionBench-VQA.** Illustrative samples from different regions are shown on the left. The map on the right depicts the regional distribution of images across India, with counts per State/UT. Further details are provided in Section D of the Appendix.

Commons–licensed images[2] from Google Search, targeting roughly 100 per State/UT across the same categories. This yielded 3,502 additional images, bringing the total corpus to 4,117 (3,797 region-specific and 320 pan-India).

Each image was first annotated with concise keywords by humans, expanded into intermediary synthetic detailed captions using VLMs in English, and then used to generate six QAs per image: two short-answer, one long-answer, one multiple-choice (single-correct), one True/False, and one adversarial question. Notably, adversarial questions incorporate false assumptions, requiring models to explicitly reject them, enabling a systematic probe of cultural knowledge beyond surface-level recognition. We employed Gemini-1.5-Flash and Gemini-2.5-Flash (Gemini Team, Google, 2025) for QA generation, informed by a small pilot study and cost considerations (see Appendix; Table 17). Human reviewers then refined all outputs for factual accuracy and cultural alignment, resulting in a balanced set of open-ended queries that jointly test recognition, reasoning, and robustness in VLMs. Guidelines provided to annotators are detailed in Appendix E while Figure 22 shows the annotation interface.

From this pool of 4K+ images and their corresponding 6 QAs, we translated a subset into the dominant regional language using text-only Gemini call, followed by human correction, resulting in an *VQA-Indic* version. Additionally, we sampled a disjoint set of 106 images and translated them into all 10 Indian languages, creating a *VQA-Parallel* corpus to systematically study cross-lingual variation in VLMs' cultural understanding and robustness.

## 3.2 INDICVISIONBENCH-MMT

The Multimodal Machine Translation (MMT) track extends the 106 images from the *VQA-Parallel* corpus, where each English caption was translated into 10 Indic languages with access to the image context. All translations were manually annotated to preserve meaning and align with cultural nuances, resulting in a multimodal parallel dataset tailored for evaluating vision-grounded multimodal translation in medium-to-low resource Indic languages.

## 3.3 INDICVISIONBENCH-OCR

For benchmarking OCR performance, we construct a multilingual corpus from Wikisource (Foundation, 2025), a public-domain repository of digitized literary works. The corpus spans 10 Indic languages and includes both printed and handwritten styles. To ensure reliability, we restrict collec-

---

[2]https://creativecommons.org/share-your-work/cclicenses/

tion to Level-4 verified pages, which have been human-reviewed on the platform. For each page, we pair high-resolution scans (`prppageimage`) with their corresponding verified text (`pagetext`). Further implementation details are provided in Appendix B.

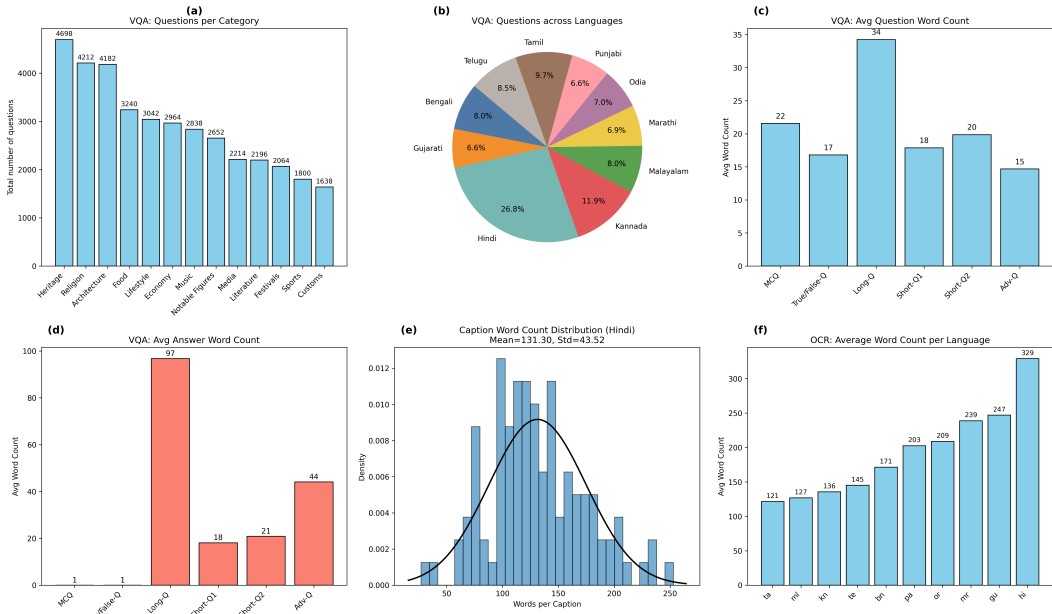

Figure 3: **Data analysis on IndicVisionBench.** Distribution of *VQA* questions by category (a) and by language excluding English (b); average word counts for questions (c) and answers (d). For *MMT* (e) shows caption word counts in Hindi; and for *OCR* average words per language (f).

## 4 INDICVISIONBENCH (IVB)

IndicVisionBench provides a diverse evaluation suite across 13 India-centric topics in English and 10 regional languages. Among Indic languages, Hindi dominates with 26.8% of QA pairs (Figure 3). For MMT, Hindi captions average 131 words, while OCR track word counts vary more widely, with Hindi (329) and Gujarati (247) highest. Figure 5 shows that the dataset spans diverse cultural categories, with largest shares in *Heritage (12.4%)*, *Religion (11.2%)*, *Architecture (11.1%)* and *Food (8.6%)*. More details in Appendix D.

**Benchmark Tracks:** IndicVisionBench consists of three evaluation subsets: *i). OCR:* 876 document images across 10 Indic languages. *ii). VQA:* 4,011 English and 1,007 multilingual culturally grounded images with 6 QA types each. We also benchmark cross-lingual performance on a disjoint set of 106 images with 6 paired questions across English and 10 Indic languages. *iii). MMT:* 106 image–caption pairs translated into 10 Indic languages, enabling multimodal translation.

**Models Evaluated.** We evaluate three families of VLMs with varying degrees of Indic language support: *i). Proprietary models:* Gemini-2.5 Flash (Gemini Team, Google, 2025), GPT-4o (OpenAI, 2023). *ii). Large open-weight VLMs:* Gemma-3-27B (Team et al., 2025), LLaMA-4-Maverick-17B (LLaMA-4 for brevity) (Meta, 2025). *iii). Medium-scale open-weight VLMs (7B):* Maya (Nahid Alam et al., 2024), PALO (Maaz et al., 2024), Pangea (Yue et al., 2025), and Chitrarth-1 (Khan et al., 2025b). For the OCR subset, we additionally include closed-source Chitrapathak[3], designed specifically for Indian languages as well as open-sourced Surya (Paruchuri & Team, 2025) model. For MMT, we additionally include Chitranuvad (Khan et al., 2025a), winning entry of the English-to-lowres[4] MMT' 24 (3 Indian languages) shared task (Parida et al., 2024).

---

[3]https://bit.ly/chitrapathak
[4]https://www2.statmt.org/wmt24/multimodallowresmt-task.html

**Evaluation Metrics** We assess model performance using a combination of deterministic and judgment-based metrics, tailored to each task. In the VQA track, Exact Match (refer Table 21) is used for multiple-choice and True/False questions, while short/long-answer and adversarial questions are evaluated using LLM-as-a-Judge (GPT-4o, 0–10 scale) following prior works (Vayani et al., 2025) to capture contextual and cultural appropriateness (prompts in Appendix F). For the MMT task, we evaluate performance using BLEU (Papineni et al., 2002) and RIBES (Isozaki et al., 2010) scores across ten Indic languages, following the setup of prior shared tasks (Parida et al., 2024). For OCR evaluation, we follow OCRBenchv2 (Fu et al., 2024b) and report Average Normalized Levenshtein Similarity (ANLS) (Biten et al., 2019), along with Word Error Rate (WER) and Character Error Rate (CER) as standard 'metrics (Smith, 2007; Neudecker et al., 2021). We report ANLS as the main metric which is more robust to outliers and also provide a detailed discussion in Section 6.

Table 1: **Model performances on English QAs in IndicVisionBench-VQA.** Average scores of different models for the six question-types. MCQ and True/False are binary (0–1), while Long Answer, Short Answer-1, Short Answer-2, and Adversarial descriptive questions use a 0–10 scale. The best score is shown in **bold**, and the second-best is underlined.

| Model | MCQ ↑ | True/False ↑ | Long-answer ↑ | Short-1 ↑ | Short-2 ↑ | Adversarial ↑ |
|---|---|---|---|---|---|---|
| Maya | 0.69 | 0.71 | 6.98 | 5.00 | 5.50 | 0.16 |
| PALO | 0.72 | 0.43 | 7.12 | 5.51 | 5.81 | 0.19 |
| Pangea | 0.85 | 0.37 | 7.01 | 6.72 | 6.95 | 0.67 |
| Chitrarth-1 | 0.81 | 0.68 | 7.53 | 6.22 | 6.33 | 0.03 |
| LLaMA-4 | 0.87 | 0.92 | 8.55 | 7.98 | 7.91 | 2.62 |
| Gemma-3 | 0.87 | 0.88 | 8.56 | 7.68 | 7.61 | 1.50 |
| GPT-4o | 0.90 | 0.91 | 8.75 | 8.19 | 8.02 | 2.95 |
| Gemini-2.5 | **0.94** | **0.95** | **9.30** | **8.58** | **8.49** | **5.79** |

## 5 RESULTS

**VQA:** Table 1 reports results on English subset of cultural VQA task. Gemini-2.5 achieves the highest scores across all 6 question types, with GPT-4o and LLaMA-4 as the strongest challengers. Binary-style questions (True/False, MCQ) yield the highest accuracy, while long-answer questions also show robust performance. Short-answer types remains harder, reflecting the difficulty of concise factual recall. This pattern highlights how answer format modulates model performance. In multilingual settings, Gemini-2.5 continues to lead overall, while LLaMA-4 and Gemma-3 exhibit comparable performance with language-specific strengths. GPT-4o consistently lags behind these models, followed by the 7B variants. Among the 7B models, Chitrarth-1 generally outperforms Pangea for short and long answer questions, with the latter holding an edge for MCQ and True/-False questions. (Figure 4; Table 8 in Appendix on VQA-Indic). Adversarial questions, which embed false assumptions, remains the most challenging both in English and Indic (Tables 1 and 2). Though Gemini-2.5 consistently outperforms all models, even its scores are notably lower compared to other QA types, reflecting the increased difficulty. On these select questions, GPT-4o is a distant second, while both Gemma-3 and LLaMA-4 struggle across the board. We also conducted paired statistical significance analyses to verify the stability of the reported model comparisons (refer to Section C.1 in the Appendix).

**MMT:** Gemini-2.5 also dominates the MMT track, with LLaMA-4 and Gemma-3 performing comparable across most languages in Table 3 based on both BLEU and RIBES metrics. LLaMA-4 attains second-best results in Bengali, Kannada, Malayalam, Odia, and Punjabi, while Gemma-3 ranks second in the remaining languages. Malayalam proves most challenging, with the sub-par performance across all models. Chitranuvad, a finetuned version of Chitrarth-1 on Visual Genome (Krishna et al., 2017) for image grounded translation of English into 3 languages (Hindi, Bengali and Malayalam), outperforms the base model Chitrarth-1 in Hindi, Kannada, Malayalam, and Telugu but lags in Bengali despite being specifically fine-tuned for it. Nevertheless, both Chitranuvad and Chitrarth-1 substantially outperform other 7B baselines (Maya, PALO, Pangea).

**OCR:** We report ANLS scores in Table 4, median WER/CER in 10. Gemini-2.5 leads across all languages and metrics, achieving SOTA performance at both the word and character-level. OCR

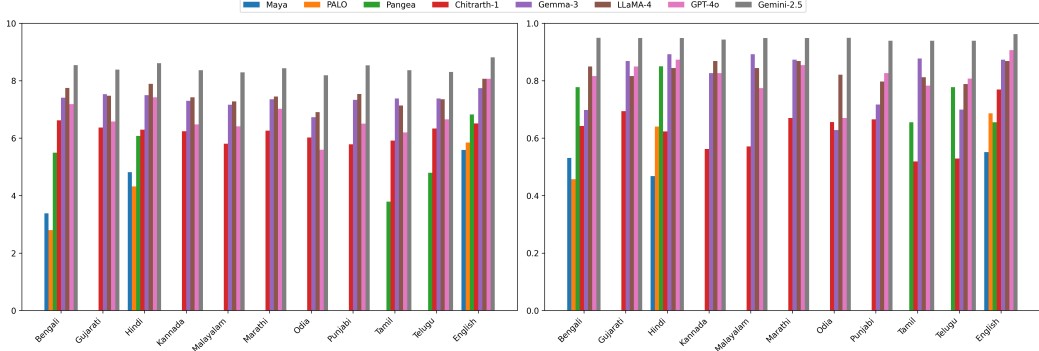

Figure 4: **Model performances on IndicVisionBench-VQA-Parallel.** Average scores across languages for the three open-ended (long and short) questions (on left) and scores across languages for the structured tasks (True/False and MCQ) on the right.

difficulty remains language-dependent, with higher scores for Malayalam (59.64), Odia (41.7), Telugu (33.32), and Gujarati (24.09), underscoring persistent challenges in Indic scripts. Surya ranks second in Gujarati, Hindi, Kannada and Marathi, while Chitrapathak ranks second in Bengali, Tamil and Telugu. For the other languages, the two models show complementary strengths, with shifts in rank depending on whether word or character level ANLS is considered. Surprisingly, GPT-4o performs poorly in OCR with word-level ANLS scores (e.g., 94.67 in Malayalam, 90.54 in Gujarati) significantly below expectations while 7B open-source models fall further behind.

Across all evaluation tracks, the closed-source Gemini-2.5 demonstrates clear superiority, while Gemma-3 and LLaMA-4 show notable strengths with observed disparities across languages and question types. We show qualitative results and more details in Appendix C.

Table 2: **Model performances for Adversarial Questions in IndicVisionBench-VQA.** We report the average scores for only top 4 models since scores of other 7B models approached to 0. Even proprietary models perform poorly on these kinds of hard and challenging questions.

| Model | Bengali ↑ | English ↑ | Gujarati ↑ | Hindi ↑ | Kannada ↑ | Malayalam ↑ | Marathi ↑ | Odia ↑ | Punjabi ↑ | Tamil ↑ | Telugu ↑ |
|---|---|---|---|---|---|---|---|---|---|---|---|
| LLaMA-4 | 0.38 | 2.62 | 0.52 | 1.18 | 0.14 | 0.33 | 0.81 | 0.53 | 1.03 | 1.14 | 0.07 |
| Gemma-3 | 1.07 | 1.50 | 0.97 | 1.66 | 1.02 | 0.77 | 0.68 | 0.90 | 2.94 | 1.85 | 1.13 |
| GPT-4o | 2.23 | 2.95 | **3.10** | 2.25 | 0.67 | 2.28 | 2.89 | 1.82 | 4.00 | 1.70 | 2.04 |
| Gemini-2.5 | **5.17** | **5.79** | 2.94 | **4.46** | **3.17** | **3.32** | **4.84** | **3.92** | **5.71** | **5.15** | **2.73** |

Table 3: **Model performances on IndicVisionBench-MMT.** RIBES (R) and BLEU (B) scores across ten Indic languages, with Gemini-2.5 achieving the highest performance consistently.

| Model | Bengali | | Gujarati | | Hindi | | Kannada | | Malayalam | | Marathi | | Odia | | Punjabi | | Tamil | | Telugu | |
|---|---|---|---|---|---|---|---|---|---|---|---|---|---|---|---|---|---|---|---|---|
| | R ↑ | B ↑ | R ↑ | B ↑ | R ↑ | B ↑ | R ↑ | B ↑ | R ↑ | B ↑ | R ↑ | B ↑ | R ↑ | B ↑ | R ↑ | B ↑ | R ↑ | B ↑ | R ↑ | B ↑ |
| Maya | 0.45 | 5.48 | – | – | 0.69 | 18.09 | – | – | – | – | – | – | – | – | – | – | – | – | – | – |
| PALO | 0.41 | 4.56 | – | – | 0.58 | 11.79 | – | – | – | – | – | – | – | – | – | – | – | – | – | – |
| Pangea | 0.69 | 16.84 | – | – | 0.75 | 25.29 | – | – | – | – | – | – | – | – | – | – | 0.43 | 5.40 | 0.62 | 12.52 |
| Chitrarth-1 | 0.76 | 21.89 | 0.72 | 21.07 | 0.71 | 21.93 | 0.65 | 12.83 | 0.59 | 7.49 | 0.70 | 16.25 | 0.62 | 11.10 | 0.50 | 10.39 | 0.71 | 17.59 | 0.67 | 15.60 |
| Chitranuvad | 0.74 | 18.13 | 0.68 | 18.66 | 0.74 | 21.93 | 0.69 | 12.93 | 0.60 | 7.36 | 0.69 | 14.74 | 0.03 | 0.86 | 0.07 | 1.61 | 0.67 | 15.85 | 0.71 | 16.56 |
| LLaMA-4 | 0.82 | 30.70 | 0.80 | 29.84 | 0.81 | 33.55 | 0.76 | 20.91 | 0.72 | 14.96 | 0.76 | 20.49 | 0.72 | 15.35 | 0.85 | 41.01 | 0.80 | 25.22 | 0.78 | 22.35 |
| Gemma-3 | 0.81 | 29.75 | 0.83 | 35.76 | 0.82 | 34.40 | 0.72 | 16.23 | 0.68 | 10.29 | 0.80 | 26.96 | 0.65 | 8.56 | 0.81 | 32.48 | 0.82 | 29.97 | 0.82 | 31.35 |
| GPT-4o | 0.80 | 28.65 | 0.74 | 21.99 | 0.79 | 33.30 | 0.67 | 11.75 | 0.59 | 8.08 | 0.75 | 23.19 | 0.65 | 9.42 | 0.75 | 24.72 | 0.73 | 16.77 | 0.71 | 17.65 |
| Gemini-2.5 | **0.87** | **44.51** | **0.90** | **53.27** | **0.83** | **38.91** | **0.80** | **30.08** | **0.81** | **28.65** | **0.88** | **47.00** | **0.85** | **39.08** | **0.89** | **52.39** | **0.88** | **46.32** | **0.87** | **44.85** |

## 6 DISCUSSION

**VLMs without vision: Are images necessary?** We evaluate models on the paired *VQA-Parallel* corpus spanning 10 Indic languages plus English, comparing performance with and without visual input. Removing images leads to a substantial drop in accuracy, most pronounced for short-answer tasks (see Table 5) where precise, detail-oriented responses are required. Long-answer questions are comparatively more resilient, though still affected. Across the representative models of each category: Chitrarth-1, Gemma-3, and Gemini-2.5, the trend is consistent (Table 9 in Appendix), showcasing that visual grounding is necessary for answering questions in our VQA benchmark.

Table 4: **Model performances on IndicVisionBench-OCR**: ANLS (Average Normalized Levenshtein Similarity) across 10 Indic languages for different models. ANLS-W and ANLS-C denote word- and character-level scores, respectively. For each language, the highest score is marked in **bold**, while the second-highest is underlined. Gemini-2.5 performs the best followed by Chitrapathak in most languages.

| Model | Bengali | | Gujarati | | Hindi | | Kannada | | Malayalam | | Marathi | | Odia | | Punjabi | | Tamil | | Telugu | |
|---|---|---|---|---|---|---|---|---|---|---|---|---|---|---|---|---|---|---|---|---|
| | Word↓ | Char↓ | Word↓ | Char↓ | Word↓ | Char↓ | Word↓ | Char↓ | Word↓ | Char↓ | Word↓ | Char↓ | Word↓ | Char↓ | Word↓ | Char↓ | Word↓ | Char↓ | Word↓ | Char↓ |
| Maya | 99.42 | 95.77 | - | - | 99.70 | 94.91 | - | - | - | - | - | - | - | - | - | - | - | - | - | - |
| PALO | 96.30 | 91.15 | - | - | 99.26 | 91.98 | - | - | - | - | - | - | - | - | - | - | - | - | - | - |
| Pangea | 94.66 | 80.33 | - | - | 99.53 | 91.50 | - | - | - | - | - | - | - | - | - | - | 99.44 | 84.13 | 99.95 | 89.91 |
| Chitrarth-1 | 96.16 | 84.65 | 99.32 | 86.81 | 98.56 | 89.81 | 99.58 | 85.29 | 99.62 | 94.77 | 99.66 | 86.58 | 99.99 | 93.21 | 99.16 | 90.17 | 99.10 | 89.94 | 99.86 | 89.02 |
| LLaMA-4 | 31.52 | 13.21 | 40.56 | 18.38 | 25.73 | 11.91 | 36.90 | 11.17 | 75.50 | 45.75 | 20.94 | 8.05 | 97.51 | 86.78 | 29.77 | 12.68 | 31.36 | 10.79 | 57.07 | 18.72 |
| Gemma-3 | 42.15 | 24.41 | 60.07 | 38.49 | 46.47 | 29.50 | 84.22 | 54.24 | 92.06 | 72.64 | 50.40 | 31.06 | 92.67 | 70.72 | 70.88 | 42.65 | 39.52 | 16.51 | 86.76 | 54.14 |
| Chitrapathak | 17.14 | 7.03 | 49.99 | 27.80 | 25.55 | 13.74 | 26.24 | 8.78 | 71.97 | 48.19 | 15.68 | 6.09 | 50.72 | 31.62 | 17.70 | 7.87 | 19.25 | 5.81 | 38.79 | 11.00 |
| Surya | 28.76 | 12.61 | 33.33 | 12.96 | 20.11 | 8.38 | 24.4 | 6.37 | 73.46 | 36.37 | 13.41 | 4.33 | 52.7 | 25.58 | 18 | 7.31 | 25.75 | 7.71 | 51.72 | 16.85 |
| GPT-4o | 55.51 | 32.68 | 90.54 | 68.03 | 54.62 | 35.54 | 94.33 | 69.79 | 94.67 | 78.47 | 63.44 | 37.93 | 94.61 | 73.46 | 68.88 | 40.71 | 74.35 | 43.39 | 95.97 | 70.08 |
| Gemini-2.5 | **11.30** | **4.04** | **24.09** | **7.61** | **16.01** | **5.88** | **17.18** | **4.38** | **59.64** | **30.60** | **8.06** | **1.79** | **41.70** | **18.60** | **14.56** | **4.98** | **15.26** | **3.01** | **33.32** | **7.16** |

Table 5: **Are images necessary for IndicVisionBench-VQA-Parallel?** Average performance drop in short-answer questions across languages for Chitrarth-1, Gemma-3, and Gemini-2.5, comparing with vs. without image input.

| Model | Type | Bengali ↑ | English ↑ | Gujarati ↑ | Hindi ↑ | Kannada ↑ | Malayalam ↑ | Marathi ↑ | Odia ↑ | Punjabi ↑ | Tamil ↑ | Telugu ↑ |
|---|---|---|---|---|---|---|---|---|---|---|---|---|
| Chitrarth-1 | w/o img | 3.88 | 4.18 | 3.76 | 4.09 | 4.07 | 3.99 | 4.53 | 4.06 | 4.52 | 3.88 | 4.23 |
| | with img | 5.90 | 5.95 | 5.76 | 5.97 | 5.58 | 4.68 | 5.61 | 5.11 | 5.43 | 4.93 | 5.50 |
| Gemma-3 | w/o img | 4.21 | 3.25 | 4.30 | 4.47 | 3.90 | 4.23 | 4.31 | 3.54 | 4.15 | 4.26 | 4.44 |
| | with img | 6.67 | 6.98 | 7.08 | 6.87 | 6.29 | 6.41 | 6.58 | 5.94 | 6.80 | 6.92 | 6.93 |
| Gemini-2.5 | w/o img | 4.69 | 4.14 | 4.62 | 4.76 | 4.29 | 4.69 | 4.57 | 4.80 | 4.60 | 4.24 | 4.66 |
| | with img | 8.09 | 8.22 | 7.90 | 8.33 | 7.57 | 7.89 | 7.99 | 7.72 | 8.15 | 7.96 | 7.76 |

**Do VLMs exhibit cross-lingual variations in performance?** We systematically conducted a study on the *VQA-Parallel* corpus to measure the cross-lingual performance across 11 languages including English. For long-answer questions, Gemini-2.5 attains the highest overall performance, followed by Gemma-3 and LLaMA-4. On the MCQ type questions, GPT-4o and Gemma-3 perform comparable across all languages (next to Gemini-2.5) as in Table 7. In the adversarial question, Gemini remains the best-performing model but shows a considerable drop in performance compared to its long-form results. Excluding English, Gemma-3 consistently outperforms both GPT-4o and LLaMA-4 across all Indian languages, while LLaMA-4 maintains a slight advantage over Gemma-3 in English. For short-answer questions, LLaMA-4 generally secures the second rank, but falls behind Gemma-3 in Tamil and Telugu. Nonetheless, both LLaMA-4 and Gemma-3 outperform GPT-4o across all languages. In the True/False setting, GPT-4o ranks second in Punjabi, Telugu, and English. By contrast, Gemma-3 shows notable weaknesses in Bengali, Punjabi, and Telugu, even trailing behind Chitrarth-1 in Punjabi and Bengali.

**Do VLMs perform better in some cultural topics in English?** Model performance also varies by cultural category. As shown in Figure 5, Gemini-2.5 consistently achieves the strongest results across topics with slight variations, establishing itself as the most reliable model. LLaMA-4 and Gemma-3 show advantages on certain topics, while GPT-4o retains a slight edge in others over both. Among 7B models, Chitrarth-1 and Pangea demonstrate moderate and roughly comparable capabilities, whereas Maya and PALO cluster together at the lower end. These topics-level patterns suggest that stronger models generalize more evenly across cultural domains, while weaker ones exhibit sharper inconsistencies.

**Do VLMs know more about certain regions or show regional-language biases?** We investigate whether VLMs exhibit region-specific strengths or biases by comparing performance across cultural images from different Indian regions (states) and the corresponding multilingual queries. Gemini-2.5 (Figure 6) generally performs well across regions but consistently struggles with Odia cultural content, regardless of query language. Across models, English questions yield the best results, followed by Hindi and Bengali, with no clear alignment between the region depicted in the image and the language of the query. Among open-weight models (Figure 7 in Appendix), Gemma-3 favors Punjabi, Marathi, and Bengali, while LLaMA-4 performs best on Punjabi and Hindi content. 7B Chitrarth-1 records its lowest scores on Odia and Punjabi and often performs better in English than in native Indic languages for Hindi-speaking states. Pangea performs strongest in English and weakest in Tamil. Maya and PALO remain relatively stable in English but show weaknesses in Hindi and Bengali. These results suggest that while certain region–language preferences exist, systematic

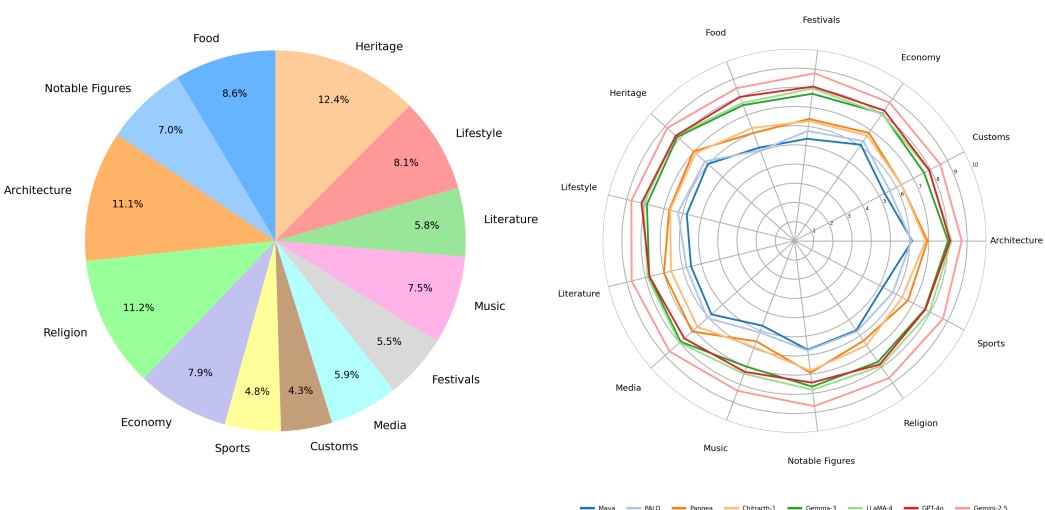

Figure 5: **Performance across topics in IndicVisionBench-VQA.** Distribution of categories of questions (on left) and model performances averaged over the two short and a long answer open-ended questions (on right). Gemini-2.5 shows comparable performance across all topics.

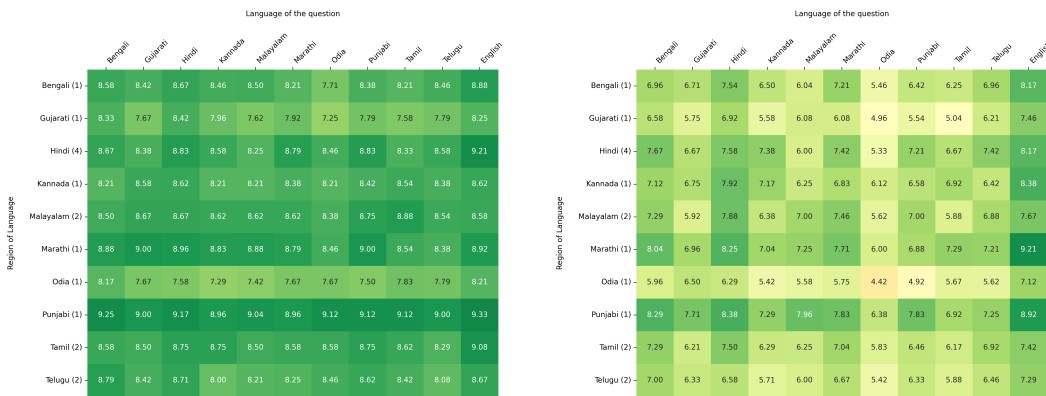

Figure 6: **Average performance on open-ended question (Long and Short answer types).** Gemini-2.5 (on left), GPT-4o (right) and other models in Figure 7 in Appendix. X-axis displays query languages and Y-axis displays Indian states grouped by dominant language. Numbers in parentheses indicate the count of states for dominant language.

region-level cultural alignment is largely absent, with Odia emerging as a consistently difficult case across all models.

**How do we evaluate OCR outputs?** Apart from the ANLS metric, we also report average and median WER/CER metrics (Tables 11 and 12 in the Appendix), based on Levenshtein Distance (Lcvenshtcin, 1966). Mathematically, this metric is unbounded and can over-penalize models for a few extreme cases (e.g., inflated scores for LLaMA-4 repetition upto maximum length; see Figure 8). To quantify this effect, we report in Table 13, the proportion of instances exceeding a value of 1. Notably, LLaMA-4 accounts for only 7% of such cases, yet includes strong outliers with an average worst-case WER of 25 in Malayalam, while still ranking third best under ANLS which remains relatively robust to these anomalies. While underexplored for LLMs and VLMs, that often produce long repetitive outputs (Hiraoka & Inui, 2024), even median-based reporting (Patel et al., 2025) fails to capture such edge cases. Other statistical alternatives like Word Recognition Rate (WRR) / Character Recognition Rate (CRR) (Bhattacharyya et al., 2025) ignore ordering in the outputs, so we adopt ANLS as the most interpretable metric in our setting.

## 7 CONCLUSION

We present IndicVisionBench, a large-scale benchmark consisting of 5K unique images, 37K+ questions spanning 13 culturally grounded topics across English and 10 Indic languages. Covering VQA (6 kind of questions), OCR, and MMT tasks, it combines curated images with linguistically diverse queries to probe recognition, reasoning, and translation. Experiments with proprietary and open-weight models reveal substantial performance gaps, especially in low-resource languages and culturally nuanced settings. By centering cultural and linguistic diversity, our work provides a reproducible foundation for building more inclusive and globally robust multimodal systems.

## ETHICS AND REPRODUCIBILITY STATEMENT

**Ethics Statement** This work focuses on the responsible development of an evaluation benchmark for multimodal cultural understanding in regional Indian contexts and languages, spanning diverse tasks. We applied careful filtering to reduce harmful or unsafe content, though model outputs remain beyond our full control. All external datasets and tools are properly cited. Human involvement was limited to annotation and quality control; no sensitive or personally identifiable information (PII) was collected. Participants were informed that their contributed images and annotations would be used in a VLM benchmark and provided prior consent, with instructions to obscure any identifiable information. Dataset curation was performed by a team of in-house annotators who were fairly compensated according to local market standards. As the study did not involve personal or medical data, formal IRB approval was not required. Throughout the process, we prioritized preserving cultural nuance while minimizing bias and harm. Despite careful filtering, dataset bias may remain, reflecting regional, socio-economic, or cultural imbalances. The resulting benchmark aims to support the development of multilingual and culturally inclusive vision-language models.

**Reproducibility Statement** To support reproducibility, we have released all benchmark-related artifacts publicly, along with detailed documentation. Our experimental setups and evaluation protocols are thoroughly recorded to facilitate precise replication of results. For components involving human annotation or judgment, we include the instructions and guidelines followed, ensuring transparency and consistency throughout the process.

## ACKNOWLEDGMENTS

We express our sincere gratitude to the leadership at Krutrim for their unwavering support throughout the course of this research. We would also like to thank the AI Research team at Krutrim for their valuable feedback and insightful discussions during various stages of the project, as well as the Krutrim team members who contributed to data collection. We further acknowledge the dedicated efforts of the data collection and annotation teams, including Sanmathi and Aravind, for their work in building this benchmark. Our experiments were conducted with generous computational support from Krutrim Cloud using Krutrim credits.

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

APPENDIX

## A    LIMITATIONS

IndicVisionBench covers English and ten medium- to low-resource Indic languages across 13 culturally grounded topics, but some limitations remain. Language coverage is still limited for some of the lowest-resource languages, and topic diversity could be further expanded to cover additional cultural contexts. Human annotations and translations are usually subject to interpretation, especially for cultural nuances, which may introduce inconsistencies or simplifications. Additionally, since parts of the dataset were seeded using LLM-generated drafts, even though these were extensively rewritten by annotators, a small degree of model-induced *self-enhancement / preferential* bias may still persist. We also acknowledge that existing evaluation metrics and LLM-based scoring may not fully capture cultural grounding or multimodal reasoning, highlighting opportunities for more nuanced evaluation approaches. Although we systematically study the impact of visual modality for VQA, we have not yet explored this effect for the MMT track with prior works (Grönroos et al., 2018; Lala et al., 2018; Wu et al., 2021) showcasing minimal impact of visual modality. We plan to cover this as part of future work.

## B    IMPLEMENTATION

Most of our implementation is based in Python and we use the HuggingFace library in PyTorch (Paszke et al., 2019; Wolf et al., 2019). We evaluate a broad range of frontier and open-source models on the IndicVisionBench benchmark using their respective APIs as well as HuggingFace pages. For the OCR-focused subset of our evaluation, we used the implementation of OCRBenchV2[5] to compute the ANLS metrics. We adapted this script to report both word-level and character-level ANLS metrics. Beyond OCR, our full benchmark includes additional evaluation dimensions. We compute metrics such as BLEU[6], where we do not have an additional pre-processing step of tokenization, RIBES[7], Exact Match (EM), and employ LLM-as-a-Judge evaluation to capture semantic and task-level correctness. We use official Batch APIs to get the results from GPT-4o[8] for LLM-as-a-judge[9] results and Gemini-2.5 for annotations.

**OCR benchmark.** We began with raw dumps from Wikisource[10] for ten Indic languages, obtained from the official Wikimedia snapshots [11]. Each compressed XML dump was parsed to extract page titles, which were then converted into canonical Wikisource URLs. From the harvested URLs, we fetched the corresponding Wikisource pages, retained only pages with images and applied filtering to retain only those marked as proofread by the community (quality level 4), ensuring high-fidelity ground truth. For every verified page, we used the <prp-page-image> tag to collect the page scans and the <pagetext> tag to extract the corresponding OCR text, which reflects the latest human-edited annotation. This pipeline, implemented with custom parsing code and filtering, yielded a linguistically diverse dataset of high-quality scanned documents paired with verified text, which forms the foundation of the OCR evaluation track in IndicVisionBench.

---

[5]https://github.com/Yuliang-Liu/MultimodalOCR/blob/main/OCRBench_v2/eval_scripts/vqa_metric.py
[6]https://huggingface.co/spaces/evaluate-metric/sacrebleu/blob/main/sacrebleu.py
[7]https://github.com/nttcslab-nlp/RIBES
[8]https://platform.openai.com/docs/guides/batch
[9]GPT-4o-2024-08-06 is the model used as LLM-as-a-Judge.
[10]https://hi.wikisource.org
[11]https://dumps.wikimedia.org/hiwikisource/latest/

## C ADDITIONAL RESULTS

We would like to highlight that IVB-VQA consists of both web scraped as well as crowdsourced images. These user-contributed images are not indexed on the web and thus do not appear in any public pretraining corpus directly. On this fully unseen subset, Gemini-2.5 again achieves the highest performance, mirroring the overall trend (see Table 6).

Table 6: **Model performances on English QAs on the subset containing crowd-sourced images of IndicVisionBench-VQA.** Average scores of different models for the four open-ended question-types: Long Answer, Short Answer-1, Short Answer-2, and Adversarial. The scores are on a scale of 0-10. The best score is shown in **bold**, and the second-best is underlined. We find a similar trend as the whole VQA dataset.

| Model | Long-answer ↑ | Short-1 ↑ | Short-2 ↑ | Adversarial ↑ |
|---|---|---|---|---|
| Maya | 6.80 | 5.28 | 6.45 | 0.06 |
| PALO | 6.98 | 5.35 | 6.51 | 0.06 |
| Pangea | 6.42 | 6.76 | 7.48 | 0.41 |
| Chitrarth-1 | 7.54 | 6.79 | 7.28 | 0.04 |
| LLaMA-4 | 8.64 | 8.43 | 8.45 | 1.56 |
| Gemma-3 | 8.74 | 8.47 | 8.45 | 0.99 |
| GPT-4o | 8.78 | 8.40 | 8.31 | 0.28 |
| Gemini-2.5 | **9.50** | **9.27** | **9.09** | **4.84** |

Table 7: **Model performances on IndicVisionBench-VQA-Parallel.** Comprehensive scores across languages, question types, and models for IndicVisionBench-VQA-Parallel.

| Q-Type | Model | Bengali | English | Gujarati | Hindi | Kannada | Malayalam | Marathi | Odia | Punjabi | Tamil | Telugu |
|---|---|---|---|---|---|---|---|---|---|---|---|---|
| MCQ (Exact Match, out of 1, ↑) | Maya | 0.462 | 0.632 | - | 0.575 | - | - | - | - | - | - | - |
| | PALO | 0.604 | 0.802 | - | 0.660 | - | - | - | - | - | - | - |
| | Pangea | 0.783 | 0.840 | - | 0.849 | - | - | - | - | - | 0.670 | 0.764 |
| | Chitrarth-1 | 0.726 | 0.811 | 0.755 | 0.792 | 0.651 | 0.726 | 0.774 | 0.679 | 0.726 | 0.689 | 0.708 |
| | LLaMA-4 | 0.802 | 0.858 | 0.802 | 0.792 | 0.840 | 0.830 | 0.811 | 0.774 | 0.802 | 0.802 | 0.783 |
| | Gemma-3 | 0.849 | 0.877 | 0.877 | 0.877 | 0.868 | 0.858 | 0.849 | 0.830 | 0.849 | 0.858 | 0.840 |
| | GPT-4o | 0.830 | 0.896 | 0.849 | 0.877 | 0.830 | 0.745 | 0.830 | 0.708 | 0.774 | 0.755 | 0.821 |
| | Gemini | **0.925** | **0.943** | **0.953** | **0.953** | **0.943** | **0.953** | **0.943** | **0.972** | **0.962** | **0.925** | **0.953** |
| True/False (Exact Match, out of 1, ↑) | Maya | 0.600 | 0.470 | - | 0.360 | - | - | - | - | - | - | - |
| | PALO | 0.310 | 0.570 | - | 0.620 | - | - | - | - | - | - | - |
| | Pangea | 0.770 | 0.470 | - | 0.850 | - | - | - | - | - | 0.640 | 0.790 |
| | Chitrarth-1 | 0.560 | 0.730 | 0.630 | 0.450 | 0.470 | 0.420 | 0.566 | 0.632 | 0.604 | 0.349 | 0.349 |
| | LLaMA-4 | 0.896 | 0.877 | 0.830 | 0.896 | 0.896 | 0.858 | 0.925 | 0.868 | 0.792 | 0.821 | 0.792 |
| | Gemma-3 | 0.547 | 0.868 | 0.858 | 0.906 | 0.783 | 0.925 | 0.896 | 0.425 | 0.585 | 0.896 | 0.557 |
| | GPT-4o | 0.802 | 0.915 | 0.849 | 0.868 | 0.821 | 0.802 | 0.877 | 0.632 | 0.877 | 0.811 | 0.792 |
| | Gemini | **0.972** | **0.981** | **0.943** | **0.943** | **0.943** | **0.943** | **0.953** | **0.925** | **0.915** | **0.953** | **0.925** |
| Long answer (LLM-as-Judge, out of 10, ↑) | Maya | 3.538 | 6.915 | - | 6.217 | - | - | - | - | - | - | - |
| | PALO | 2.557 | 7.057 | - | 5.217 | - | - | - | - | - | - | - |
| | Pangea | 6.585 | 7.038 | - | 7.009 | - | - | - | - | - | 5.066 | 5.887 |
| | Chitrarth-1 | 7.443 | 7.491 | 7.547 | 7.311 | 7.292 | 7.406 | 7.472 | 7.311 | 6.972 | 7.142 | 7.443 |
| | LLaMA-4 | 8.396 | 8.566 | 8.217 | 8.500 | 8.387 | 7.934 | 8.349 | 7.774 | 8.292 | 8.236 | 8.292 |
| | Gemma-3 | 8.377 | 8.698 | 8.358 | 8.443 | 8.377 | 8.104 | 8.368 | 7.802 | 8.330 | 8.443 | 8.236 |
| | GPT-4o | 8.075 | 8.660 | 7.868 | 8.330 | 7.613 | 7.557 | 8.170 | 6.868 | 7.642 | 7.528 | 7.764 |
| | Gemini | **9.094** | **9.453** | **9.113** | **9.132** | **9.113** | **8.877** | **9.075** | **8.764** | **9.142** | **8.981** | **8.981** |
| Short-answer 1 (LLM-as-Judge, out of 10, ↑) | Maya | 3.142 | 4.745 | - | 3.755 | - | - | - | - | - | - | - |
| | PALO | 3.066 | 5.000 | - | 3.708 | - | - | - | - | - | - | - |
| | Pangea | 4.557 | 6.170 | - | 5.443 | - | - | - | - | - | 3.066 | 4.094 |
| | Chitrarth-1 | 5.896 | 5.953 | 5.755 | 5.972 | 5.575 | 4.679 | 5.613 | 5.114 | 5.434 | 4.925 | 5.500 |
| | LLaMA-4 | 7.198 | 7.387 | 7.189 | 7.415 | 6.698 | 6.736 | 6.868 | 6.283 | 6.991 | 6.302 | 6.679 |
| | Gemma-3 | 6.670 | 6.981 | 7.075 | 6.868 | 6.292 | 6.406 | 6.575 | 5.943 | 6.802 | 6.915 | 6.934 |
| | GPT-4o | 6.726 | 7.594 | 6.028 | 7.075 | 5.896 | 5.962 | 6.519 | 4.849 | 6.019 | 5.538 | 6.123 |
| | Gemini | **8.094** | **8.217** | **7.896** | **8.330** | **7.566** | **7.887** | **7.991** | **7.717** | **8.151** | **7.962** | **7.755** |
| Short-answer 2 (LLM-as-Judge, out of 10, ↑) | Maya | 3.462 | 5.094 | - | 4.472 | - | - | - | - | - | - | - |
| | PALO | 2.774 | 5.472 | - | 4.028 | - | - | - | - | - | - | - |
| | Pangea | 5.340 | 7.255 | - | 5.783 | - | - | - | - | - | 3.236 | 4.396 |
| | Chitrarth-1 | 6.519 | 6.085 | 5.792 | 5.604 | 5.849 | 5.330 | 5.698 | 5.651 | 4.953 | 5.670 | 6.066 |
| | LLaMA-4 | 7.642 | 8.236 | 7.019 | 7.755 | 7.179 | 7.151 | 7.132 | 6.651 | 7.321 | 6.858 | 7.085 |
| | Gemma-3 | 7.170 | 7.547 | 7.160 | 7.179 | 7.217 | 6.981 | 7.123 | 6.443 | 6.858 | 6.783 | 6.962 |
| | GPT-4o | 6.755 | 7.934 | 5.840 | 6.858 | 6.858 | 5.708 | 6.368 | 5.075 | 5.858 | 5.538 | 6.075 |
| | Gemini | **8.434** | **8.755** | **8.142** | **8.368** | **8.406** | **8.094** | **8.236** | **8.085** | **8.302** | **8.151** | **8.179** |
| Adversarial question (LLM-as-Judge, out of 10, ↑) | Maya | 0.255 | 0.368 | - | 0.377 | - | - | - | - | - | - | - |
| | PALO | 0.123 | 0.453 | - | 0.104 | - | - | - | - | - | - | - |
| | Pangea | 0.066 | 0.858 | - | 0.000 | - | - | - | - | - | 0.000 | 0.000 |
| | Chitrarth-1 | 0.000 | 0.094 | 0.094 | 0.075 | 0.000 | 0.000 | 0.094 | 0.000 | 0.000 | 0.047 | 0.047 |
| | LLaMA-4 | 1.123 | 3.387 | 0.849 | 1.500 | 0.953 | 1.547 | 0.821 | 0.406 | 0.991 | 0.849 | 0.660 |
| | Gemma-3 | 1.566 | 2.179 | 1.915 | 2.349 | 2.283 | 2.226 | 1.915 | 2.132 | 2.472 | 2.085 | 2.906 |
| | GPT-4o | 0.642 | 1.104 | 0.708 | 0.745 | 0.726 | 0.425 | 0.642 | 0.642 | 0.566 | 0.623 | 0.726 |
| | Gemini | **4.745** | **6.142** | **4.962** | **5.528** | **4.491** | **4.991** | **4.575** | **4.094** | **4.660** | **4.670** | **5.019** |

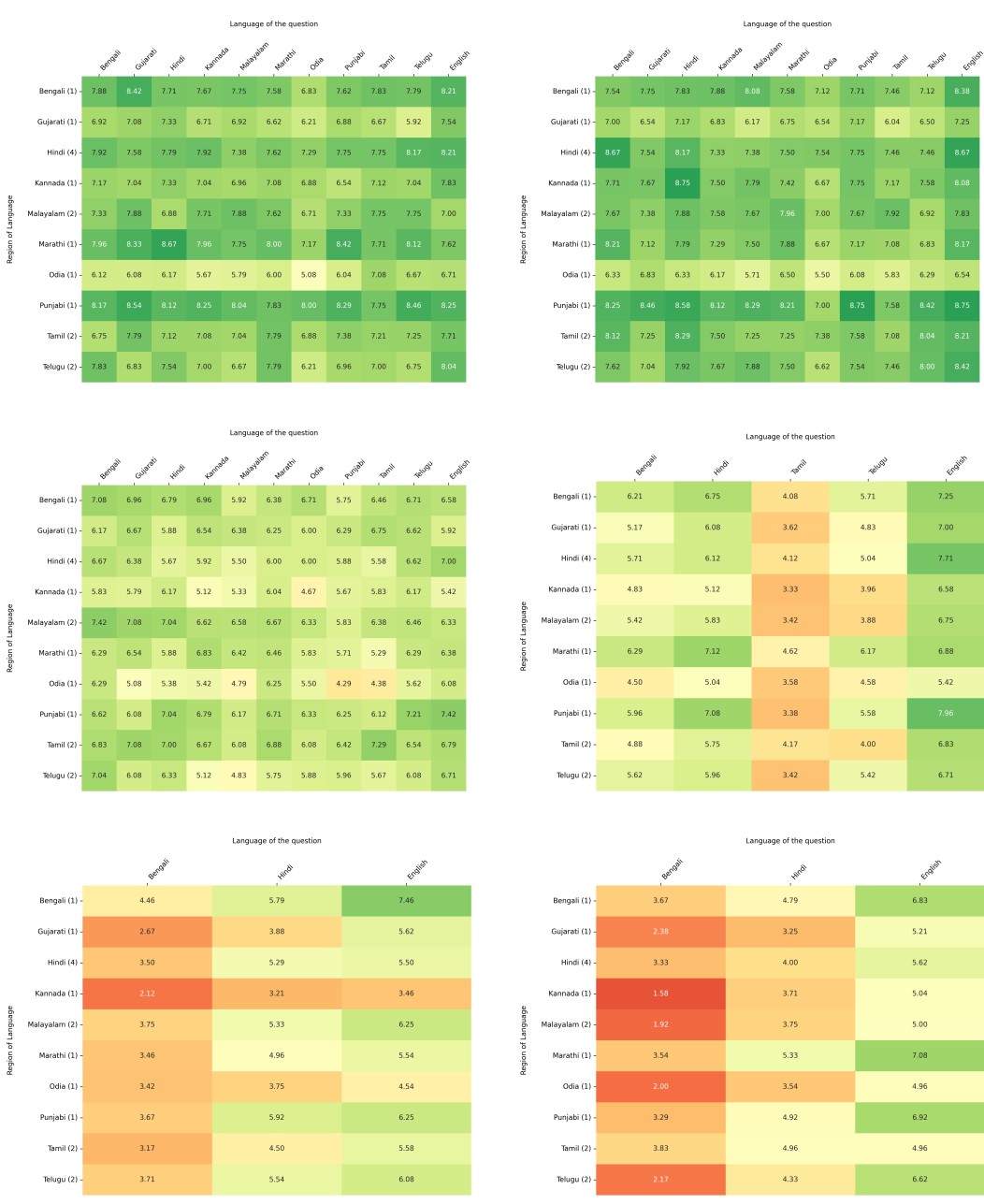

Figure 7: **Model performances on IndicVisionBench-VQA-Parallel.** Average scores on the three open-ended questions (Long and two Short) for six models across different languages (X-axis) and images corresponding to Indian states grouped by primary language (Y-axis). Left to right, top row: Gemma-3-27B (left) and LLaMA-4 (right); middle row: Chitrarth-1 (left) and Pangea (right); bottom row: Maya (left) and PALO (right).

Table 8: **Model performances on IndicVisionBench-VQA-Indic.** Comprehensive scores across languages, question types, and models.

| Q-Type | Model | Bengali | Gujarati | Hindi | Kannada | Malayalam | Marathi | Odia | Punjabi | Tamil | Telugu |
|---|---|---|---|---|---|---|---|---|---|---|---|
| MCQ (Exact Match, out of 1, ↑) | Maya | 0.433 | - | 0.638 | - | - | - | - | - | - | - |
| | PALO | 0.467 | - | 0.576 | - | - | - | - | - | - | - |
| | Pangea | 0.733 | - | 0.812 | - | - | - | - | - | 0.670 | 0.768 |
| | Chitrarth-1 | 0.717 | 0.774 | 0.812 | 0.633 | 0.733 | 0.811 | 0.605 | 0.645 | 0.691 | 0.681 |
| | LLaMA-4 | 0.700 | **0.871** | 0.866 | 0.878 | 0.817 | 0.838 | 0.816 | 0.839 | 0.809 | 0.899 |
| | Gemma-3 | 0.817 | 0.839 | 0.844 | 0.878 | 0.783 | 0.919 | 0.868 | 0.710 | 0.798 | 0.870 |
| | GPT-4o | 0.750 | 0.613 | 0.839 | 0.719 | 0.733 | 0.838 | 0.684 | 0.806 | 0.713 | 0.812 |
| | Gemini-2.5 | **0.883** | **0.871** | **0.924** | **0.906** | **0.833** | **1.000** | **0.895** | **0.935** | **0.883** | **0.928** |
| True/False (Exact Match out of 1, ↑) | Maya | 0.483 | - | 0.333 | - | - | - | - | - | - | - |
| | PALO | 0.317 | - | 0.714 | - | - | - | - | - | - | - |
| | Pangea | 0.717 | - | 0.746 | - | - | - | - | - | 0.585 | 0.725 |
| | Chitrarth-1 | 0.600 | 0.452 | 0.415 | 0.374 | 0.500 | 0.514 | 0.553 | 0.548 | 0.277 | 0.377 |
| | LLaMA-4 | 0.850 | 0.742 | 0.891 | **0.899** | 0.783 | 0.892 | 0.816 | 0.839 | 0.862 | 0.870 |
| | Gemma-3 | 0.867 | 0.581 | 0.830 | 0.827 | 0.800 | 0.757 | 0.711 | 0.935 | 0.766 | 0.812 |
| | GPT-4o | 0.883 | 0.742 | 0.842 | 0.676 | 0.850 | 0.892 | 0.711 | 0.968 | 0.819 | 0.739 |
| | Gemini-2.5 | **0.917** | **0.871** | **0.924** | 0.878 | **0.867** | **0.973** | **0.868** | **1.000** | **0.915** | **0.928** |
| Long answer (LLM-as-Judge, out of 10, ↑) | Maya | 3.867 | - | 6.504 | - | - | - | - | - | - | - |
| | PALO | 3.150 | - | 5.712 | - | - | - | - | - | - | - |
| | Pangea | 6.783 | - | 7.493 | - | - | - | - | - | 4.787 | 5.884 |
| | Chitrarth-1 | 7.233 | 7.484 | 7.547 | 7.669 | 7.433 | 7.351 | 7.553 | 7.290 | 7.298 | 7.290 |
| | LLaMA-4 | 8.050 | 8.290 | 8.482 | 8.489 | 8.267 | 8.486 | 7.763 | 8.452 | 8.245 | 8.000 |
| | Gemma-3 | 8.400 | 8.613 | 8.498 | 8.576 | 8.317 | 8.541 | 7.737 | 8.613 | 8.489 | 8.217 |
| | GPT-4o | 8.283 | 8.484 | 8.484 | 7.000 | 8.033 | 8.243 | 7.868 | 8.484 | 7.755 | 7.913 |
| | Gemini-2.5 | **8.883** | **9.226** | **9.087** | **9.029** | **8.817** | **9.243** | **8.737** | **8.968** | **8.904** | **8.870** |
| Short answer 1 (LLM-as-Judge, out of 10, ↑) | Maya | 2.117 | - | 4.199 | - | - | - | - | - | - | - |
| | PALO | 2.733 | - | 3.984 | - | - | - | - | - | - | - |
| | Pangea | 4.617 | - | 5.772 | - | - | - | - | - | 4.160 | 4.768 |
| | Chitrarth-1 | 5.550 | 6.129 | 6.328 | 5.964 | 4.483 | 5.595 | 6.658 | 5.129 | 6.489 | 5.551 |
| | LLaMA-4 | 7.050 | 7.677 | 7.710 | 7.662 | 6.933 | 7.351 | 6.289 | 7.516 | 7.394 | 7.043 |
| | Gemma-3 | 6.783 | 8.032 | 7.578 | 7.158 | 6.417 | 6.676 | 6.526 | 7.516 | 7.266 | 7.101 |
| | GPT-4o | 7.550 | 7.355 | 8.060 | 7.309 | 6.800 | 6.865 | 6.158 | 7.065 | 7.223 | 7.580 |
| | Gemini-2.5 | **8.117** | **8.355** | **8.540** | **8.734** | **7.600** | **8.486** | **8.263** | **8.355** | **8.245** | **8.116** |
| Short answer 2 (LLM-as-Judge, out of 10, ↑) | Maya | 3.717 | - | 4.696 | - | - | - | - | - | - | - |
| | PALO | 2.700 | - | 4.205 | - | - | - | - | - | - | - |
| | Pangea | 4.983 | - | 6.188 | - | - | - | - | - | 4.213 | 4.652 |
| | Chitrarth-1 | 5.883 | 6.613 | 6.212 | 6.029 | 5.567 | 6.378 | 6.132 | 6.032 | 6.468 | 6.087 |
| | LLaMA-4 | 6.917 | 7.581 | 7.665 | 7.633 | 6.917 | 7.838 | 6.947 | 7.742 | 6.926 | 7.217 |
| | Gemma-3 | 7.117 | 7.742 | 7.344 | 7.173 | 6.450 | 6.811 | 5.842 | 7.484 | 6.883 | 6.652 |
| | GPT-4o | 7.300 | 7.581 | 7.652 | 7.338 | 7.450 | 7.649 | 6.579 | 7.290 | 7.160 | 7.261 |
| | Gemini-2.5 | **7.950** | **8.484** | **8.268** | **8.691** | **8.083** | **8.730** | **8.289** | **8.452** | **8.117** | **8.188** |
| Adversarial (LLM-as-Judge, out of 10, ↑) | Maya | 0 | - | 0.188 | - | - | - | - | - | - | - |
| | PALO | 0.017 | - | 0.114 | - | - | - | - | - | - | - |
| | Pangea | 0 | - | 0.011 | - | - | - | - | - | 0 | 0 |
| | Chitrarth-1 | 0 | 0 | 0.011 | 0.072 | 0 | 0.135 | 0.053 | 0.161 | 0.053 | 0 |
| | LLaMA-4 | 0.383 | 0.516 | 1.179 | 0.144 | 0.333 | 0.811 | 0.526 | 1.032 | 1.138 | 0.072 |
| | Gemma-3 | 1.067 | 0.968 | 1.656 | 1.022 | 0.767 | 0.676 | 0.895 | 2.935 | 1.851 | 1.130 |
| | GPT-4o | 2.233 | 3.097 | 2.248 | 0.669 | 2.283 | 2.892 | 1.816 | 4.000 | 1.702 | 2.043 |
| | Gemini-2.5 | **5.167** | 2.935 | **4.460** | **3.165** | **3.317** | **4.838** | **3.921** | **5.710** | **5.149** | **2.725** |

Table 9: **VQA with and without image.** Average scores on long-answer type questions of Chitrarth-1, Gemma-3, and Gemini-2.5 on IVB-VQA-Parallel across 11 languages, evaluated with and without image input.

| Model | Type | Bengali ↑ | English ↑ | Gujarati ↑ | Hindi ↑ | Kannada ↑ | Malayalam ↑ | Marathi ↑ | Odia ↑ | Punjabi ↑ | Tamil ↑ | Telugu ↑ |
|---|---|---|---|---|---|---|---|---|---|---|---|---|
| Chitrarth-1 | w/o img | 6.52 | 6.37 | 6.72 | 6.62 | 6.69 | 6.67 | 6.52 | 6.39 | 6.09 | 6.78 | 6.65 |
| | with img | 7.44 | 7.49 | 7.55 | 7.31 | 7.29 | 7.41 | 7.47 | 7.31 | 6.97 | 7.14 | 7.44 |
| Gemma-3 | w/o img | 7.40 | 6.43 | 7.49 | 7.33 | 7.53 | 7.43 | 7.40 | 6.53 | 7.07 | 7.70 | 7.29 |
| | with img | 8.38 | 8.70 | 8.36 | 8.44 | 8.38 | 8.10 | 8.37 | 7.80 | 8.33 | 8.44 | 8.24 |
| Gemini-2.5 | w/o img | 8.11 | 6.58 | 8.13 | 8.21 | 8.15 | 8.15 | 8.27 | 8.06 | 8.13 | 8.11 | 8.21 |
| | with img | 9.09 | 9.45 | 9.11 | 9.13 | 9.11 | 8.88 | 9.08 | 8.76 | 9.14 | 8.98 | 8.98 |

Table 10: **Model performances on IndicVisionBench-OCR.** Median WER and CER scores across Indic languages for various models.

| Model | Bengali | | Gujarati | | Hindi | | Kannada | | Malayalam | | Marathi | | Odia | | Punjabi | | Tamil | | Telugu | |
|---|---|---|---|---|---|---|---|---|---|---|---|---|---|---|---|---|---|---|---|---|
| | WER↓ | CER↓ | WER↓ | CER↓ | WER↓ | CER↓ | WER↓ | CER↓ | WER↓ | CER↓ | WER↓ | CER↓ | WER↓ | CER↓ | WER↓ | CER↓ | WER↓ | CER↓ | WER↓ | CER↓ |
| Maya | 1.00 | 1.00 | - | - | 1.00 | 0.98 | - | - | - | - | - | - | - | - | - | - | - | - | - | - |
| PALO | 1.00 | 0.99 | - | - | 1.00 | 0.99 | - | - | - | - | - | - | - | - | - | - | - | - | - | - |
| Pangea | 1.00 | 0.85 | - | - | 1.00 | 0.95 | - | - | - | - | - | - | - | - | - | - | 1.00 | 0.87 | 1.00 | 0.93 |
| Chitrarth-1 | 1.00 | 0.96 | 1.00 | 0.96 | 0.99 | 0.97 | 1.11 | 0.95 | 1.00 | 0.99 | 1.00 | 0.95 | 1.19 | 1.00 | 1.00 | 0.97 | 1.00 | 0.97 | 1.00 | 0.96 |
| LLaMA-4 | 0.38 | 0.13 | 0.53 | 0.16 | 0.27 | 0.09 | 0.65 | 0.12 | 0.87 | 0.45 | 0.25 | 0.07 | 1.00 | 0.89 | 0.34 | 0.11 | 0.39 | 0.08 | 0.69 | 0.19 |
| Gemma-3 | 0.49 | 0.21 | 0.71 | 0.40 | 0.50 | 0.28 | 0.91 | 0.55 | 0.98 | 0.76 | 0.60 | 0.34 | 0.97 | 0.77 | 0.78 | 0.43 | 0.57 | 0.15 | 0.93 | 0.57 |
| GPT-4o | 0.65 | 0.29 | 1.00 | 0.77 | 0.60 | 0.34 | 1.09 | 0.74 | 1.00 | 0.80 | 0.75 | 0.39 | 0.97 | 0.79 | 0.78 | 0.39 | 0.92 | 0.44 | 1.17 | 0.76 |
| Gemini-2.5 | **0.25** | **0.03** | **0.33** | **0.07** | **0.23** | **0.04** | **0.24** | **0.04** | **0.63** | **0.30** | **0.23** | **0.02** | **0.55** | **0.20** | **0.24** | **0.04** | **0.39** | **0.03** | **0.45** | **0.05** |

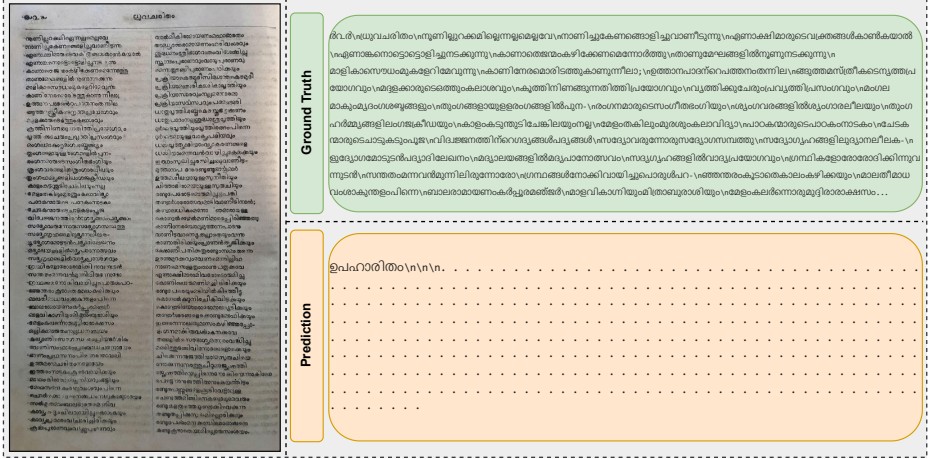

Figure 8: **Observed repetition in OCR outputs.** We show an example where LLaMA-4 provided repetitions upto maximum sequence length in the prediction for an OCR example in Malayalam.

Table 11: **Model performances on IndicVisionBench-OCR.** Average WER and CER ($\pm$ standard deviation) for Bengali, Gujarati, Hindi, Kannada, and Malayalam.

| Model | Bengali | | Gujarati | | Hindi | | Kannada | | Malayalam | |
|---|---|---|---|---|---|---|---|---|---|---|
| | WER ↓ | CER ↓ | WER ↓ | CER ↓ | WER ↓ | CER ↓ | WER ↓ | CER ↓ | WER ↓ | CER ↓ |
| Maya | 1.15 ± 0.56 | 0.99 ± 0.07 | - | - | 2.32 ± 8.94 | 1.90 ± 5.78 | - | - | - | - |
| PALO | 2.77 ± 3.45 | 2.06 ± 2.02 | - | - | 1.58 ± 2.43 | 1.06 ± 0.50 | - | - | - | - |
| Pangea | 1.25 ± 0.90 | 0.99 ± 0.59 | - | - | 1.22 ± 1.31 | 1.07 ± 1.07 | - | - | 2.45 ± 7.82 | 1.16 ± 0.61 |
| Chitrarth-1 | 1.34 ± 0.89 | 1.07 ± 0.64 | 1.38 ± 1.05 | 1.02 ± 0.48 | 1.09 ± 0.41 | 0.96 ± 0.22 | 1.37 ± 0.49 | 0.95 ± 0.13 | 2.45 ± 7.82 | 1.16 ± 0.61 |
| Chitrapathak | 0.33 ± 0.13 | 0.08 ± 0.14 | 0.55 ± 0.19 | 0.29 ± 0.25 | 0.37 ± 0.37 | 0.15 ± 0.27 | 0.34 ± 0.14 | 0.09 ± 0.08 | **0.76 ± 0.18** | 0.48 ± 0.32 |
| Gemma-3 | 0.53 ± 0.19 | 0.26 ± 0.15 | 0.71 ± 0.13 | 0.41 ± 0.13 | 0.59 ± 0.44 | 0.35 ± 0.41 | 0.94 ± 0.16 | 0.58 ± 0.15 | 1.72 ± 5.42 | 0.76 ± 0.15 |
| LLaMA-4 | 0.40 ± 0.17 | 0.14 ± 0.11 | 0.53 ± 0.28 | 0.20 ± 0.18 | 0.37 ± 0.36 | 0.14 ± 0.18 | 0.66 ± 0.29 | 0.13 ± 0.12 | 25.26 ± 217.47 | 0.48 ± 0.26 |
| GPT-4o | 0.71 ± 0.43 | 0.41 ± 0.50 | 1.36 ± 0.97 | 1.40 ± 2.51 | 0.77 ± 0.78 | 0.42 ± 0.28 | 1.43 ± 1.21 | 0.95 ± 0.72 | 7.62 ± 39.12 | 1.12 ± 0.92 |
| Gemini-2.5 | **0.26 ± 0.08** | **0.05 ± 0.09** | **0.33 ± 0.13** | **0.08 ± 0.11** | **0.29 ± 0.31** | **0.07 ± 0.12** | **0.27 ± 0.19** | **0.05 ± 0.05** | 2.26 ± 9.16 | **0.31 ± 0.26** |

Table 12: **Model performances on IndicVisionBench-OCR.** Average WER and CER ($\pm$ standard deviation) for Marathi, Odia, Punjabi, Tamil, and Telugu.

| Model | Marathi | | Odia | | Punjabi | | Tamil | | Telugu | |
|---|---|---|---|---|---|---|---|---|---|---|
| | WER ↓ | CER ↓ | WER ↓ | CER ↓ | WER ↓ | CER ↓ | WER ↓ | CER ↓ | WER ↓ | CER ↓ |
| Maya | - | - | - | - | - | - | - | - | - | - |
| PALO | - | - | - | - | - | - | - | - | - | - |
| Pangea | - | - | - | - | - | - | 1.37 ± 1.02 | 0.97 ± 0.43 | 1.29 ± 0.67 | 0.98 ± 0.35 |
| Chitrarth-1 | 1.15 ± 0.50 | 0.93 ± 0.21 | 1.89 ± 1.36 | 1.52 ± 0.93 | 1.59 ± 1.20 | 1.41 ± 1.17 | 1.53 ± 1.58 | 1.09 ± 0.56 | 1.56 ± 1.27 | 1.05 ± 0.50 |
| Chitrapathak | 0.31 ± 0.16 | 0.07 ± 0.09 | 0.60 ± 0.20 | 0.33 ± 0.27 | 0.27 ± 0.16 | 0.09 ± 0.15 | 0.43 ± 0.14 | 0.07 ± 0.12 | 0.54 ± 0.21 | 0.12 ± 0.15 |
| Gemma-3 | 0.59 ± 0.16 | 0.32 ± 0.13 | 0.98 ± 0.14 | 0.75 ± 0.12 | 0.78 ± 0.14 | 0.44 ± 0.13 | 0.65 ± 0.41 | 0.18 ± 0.12 | 1.05 ± 0.68 | 0.58 ± 0.18 |
| LLaMA-4 | 0.30 ± 0.22 | 0.09 ± 0.14 | 1.68 ± 3.58 | 1.10 ± 0.65 | 0.41 ± 0.21 | 0.13 ± 0.12 | 0.53 ± 0.50 | 0.13 ± 0.19 | 0.77 ± 0.57 | 0.20 ± 0.11 |
| GPT-4o | 0.76 ± 0.26 | 0.42 ± 0.23 | 1.26 ± 1.18 | 0.94 ± 0.75 | 0.88 ± 0.42 | 0.57 ± 0.80 | 1.23 ± 1.97 | 0.69 ± 1.63 | 1.41 ± 0.98 | 0.85 ± 0.35 |
| Gemini-2.5 | **0.25 ± 0.11** | **0.03 ± 0.02** | **0.55 ± 0.25** | **0.21 ± 0.14** | **0.25 ± 0.15** | **0.06 ± 0.11** | **0.42 ± 0.18** | **0.05 ± 0.04** | **0.51 ± 0.29** | **0.08 ± 0.10** |

Table 13: **IndicVisionBench-OCR WER and CER statistics.** Model-wise WER and CER statistics where the scores are more than 1. We present the count as well as percentage of the examples for each model.

| Model | WER > 1 | | CER > 1 | |
|---|---|---|---|---|
| | Count | % | Count | % |
| Maya | 22 | 2.51 | 15 | 1.71 |
| PALO | 51 | 5.82 | 45 | 5.14 |
| Pangea | 77 | 8.79 | 34 | 3.88 |
| Chitrarth-1 | 302 | 34.47 | 169 | 19.29 |
| Chitrapathak | 8 | 0.91 | 0 | 0.00 |
| Gemma-3 | 79 | 9.01 | 10 | 1.14 |
| LLaMA-4 | 68 | 7.76 | 28 | 3.19 |
| GPT-4o | 286 | 32.64 | 115 | 13.12 |
| Gemini-2.5 | 15 | 1.71 | 0 | 0.00 |

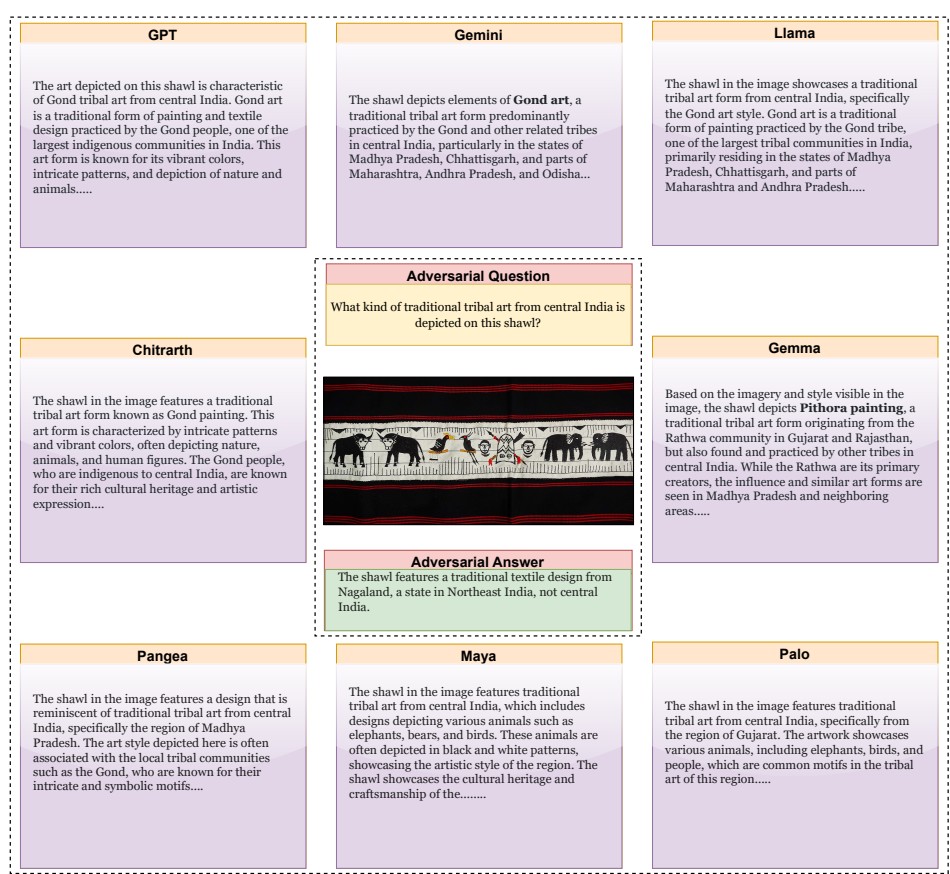

Figure 9: **Model outputs on IndicVisionBench-VQA.** We show an example of an adversarial question along with the corresponding model outputs.

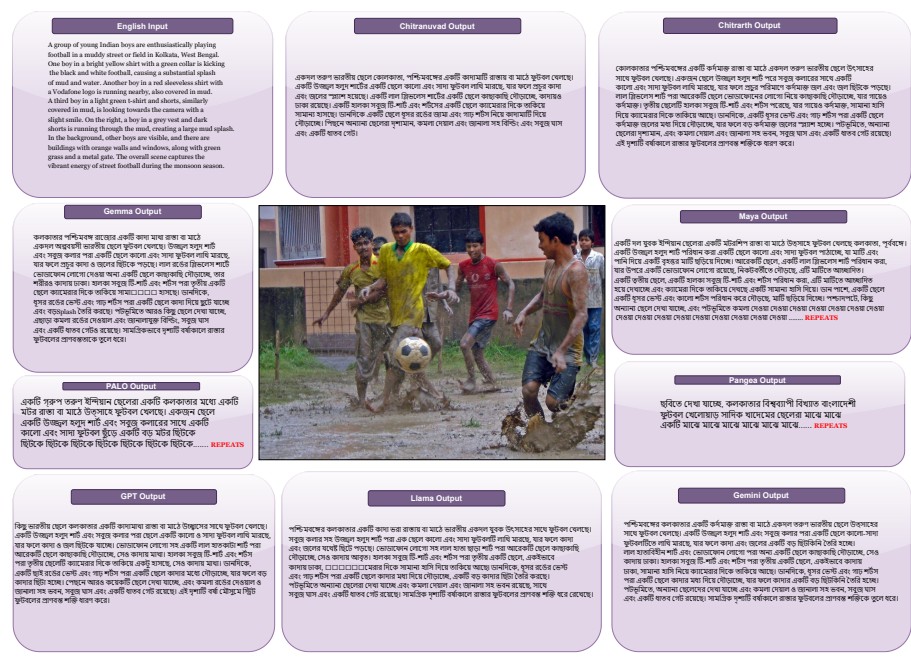

Figure 10: **Model outputs on IndicVisionBench-MMT.** Example of an MMT question and the corresponding responses from multiple models.

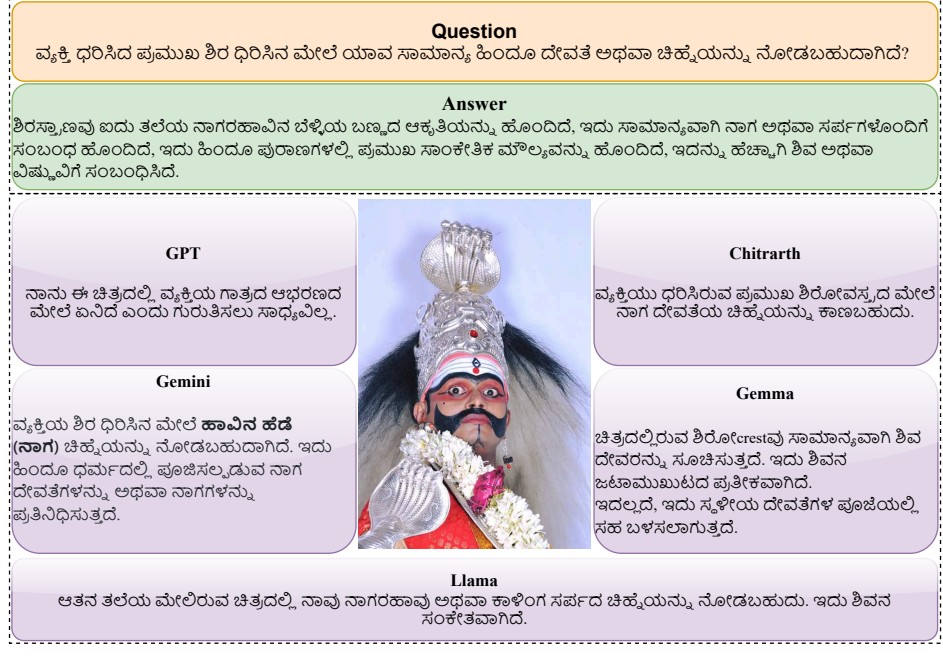

Figure 11: **Model outputs on IndicVisionBench-VQA.** We show an example of an VQA output for corresponding models.

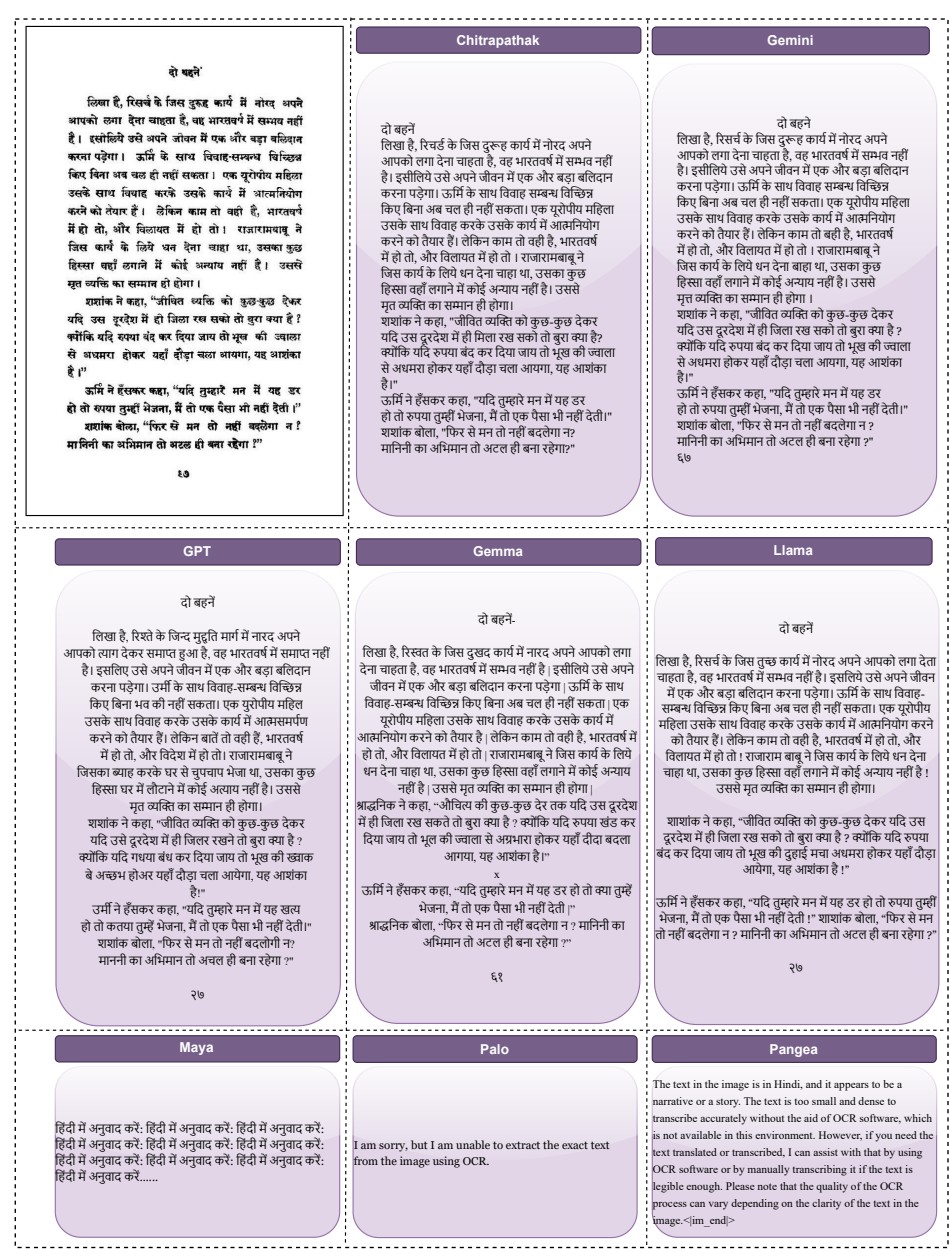

Figure 12: **Model outputs on IndicVisionBench-OCR.** We show an example of an OCR output for corresponding models.

## C.1 Statistical Analysis on the results

To assess whether Gemini provides statistically significant improvements over baseline models (GPT-4o and Gemma-3) across languages, question types, and corpora, we conduct paired statistical tests on model outputs. For each combination of corpus (VQA-Indic and VQA-Parallel), language, and QA category (Long-QA, Short-QA, and Adversarial-QA), we compute paired score differences between Gemini and each comparison model.

**Statistical Tests.** We use the Wilcoxon signed-rank test (Wilcoxon, 1945) to determine whether the distribution of paired differences differs significantly from zero.

**Confidence Intervals.** To estimate the magnitude and robustness of these differences, we also construct bootstrapped confidence intervals. Specifically, we perform 10,000 bootstrap resamples over the paired differences and compute the mean for each resample. The empirical 2.5th and 97.5th percentiles form the 95% confidence interval for the mean paired improvement.

**Findings.** Overall, Gemini performs consistently better than both GPT-4o and Gemma-3 across every language and question type we evaluated. For VQA-Indic, the statistical tests produce reliably small $p$-values, and for VQA-Parallel the values are even smaller showing that Gemini's improvements are not due to chance. The bootstrapped confidence intervals are also entirely positive, reinforcing that Gemini's advantage is both meaningful and robust (Tables 14 and 15).

The differences in statistical strength stem primarily from dataset sampling. VQA-Parallel includes a uniform 106 items per language, whereas VQA-Indic varies widely in sample size (e.g., Punjabi 31, Odia 38, Hindi 448), which naturally produces wider confidence intervals and weaker p-values for smaller languages. The only instance where significance is not observed is the Punjabi Short-QA comparison between Gemini-2.5 and Gemma-3, where the results do not provide sufficient evidence to establish a clear advantage.

Despite this expected variability, the overall results consistently show that Gemini-2.5 outperforms baseline models across languages and QA types, with improvements that are statistically significant, meaningful, and robust across both balanced and imbalanced multilingual evaluation settings.

Table 14: **Statistical comparison of models on the VQA-Indic set** across languages with p-values and confidence intervals for Long-QA, Short-QA, and Adversarial-QA.

| Model A | Model B | Language | Long-QA | | Short-QA | | Adversarial-QA | | n-items |
|---|---|---|---|---|---|---|---|---|---|
| | | | p-value | CI | p-value | CI | p-value | CI | |
| Gemini-2.5 | GPT-4o | Punjabi | 5e-4 | (0.26, 0.71) | 2e-2 | (0.29, 2.35) | 2e-2 | (0.48, 3.0) | 31 |
| Gemini-2.5 | Gemma-3 | Punjabi | 1e-2 | (0.1, 0.65) | *0.06* | *(0.0, 1.68)* | 3e-3 | (1.29, 4.26) | 31 |
| Gemini-2.5 | GPT-4o | Hindi | 1e-31 | (0.49, 0.72) | 9e-4 | (0.26, 0.7) | 9e-21 | (1.81, 2.6) | 448 |
| Gemini-2.5 | Gemma-3 | Hindi | 7e-34 | (0.5, 0.68) | 2e-12 | (0.72, 1.22) | 3e-26 | (2.36, 3.25) | 448 |
| Gemini-2.5 | GPT-4o | Odia | 1e-4 | (0.53, 1.24) | 3e-5 | (1.26, 3.03) | 1e-2 | (0.63, 3.66) | 38 |
| Gemini-2.5 | Gemma-3 | Odia | 1e-4 | (0.58, 1.47) | 1e-4 | (0.95, 2.61) | 8e-4 | (1.55, 4.61) | 38 |
| Gemini-2.5 | GPT-4o | Marathi | 1e-4 | (0.51, 1.62) | 3e-3 | (0.7, 2.57) | 2e-2 | (0.32, 3.54) | 37 |
| Gemini-2.5 | Gemma-3 | Marathi | 2e-5 | (0.46, 0.97) | 1e-3 | (0.89, 2.76) | 1e-4 | (2.73, 5.7) | 37 |

Table 15: **Statistical comparison of models on the VQA-Parallel set** across languages with p-values and confidence intervals for Long-QA, Short-QA, and Adversarial-QA.

| Model A | Model B | Language | Long-QA | | Short-QA | | Adversarial-QA | | n-items |
|---|---|---|---|---|---|---|---|---|---|
| | | | p-value | CI | p-value | CI | p-value | CI | |
| Gemini-2.5 | GPT-4o | Malayalam | 2e-14 | (1.07, 1.6) | 5e-9 | (1.33, 2.53) | 6e-12 | (3.72, 5.43) | 106 |
| Gemini-2.5 | Gemma-3 | Malayalam | 1e-8 | (0.51, 1.07) | 2e-7 | (0.94, 2.05) | 4e-7 | (1.87, 3.66) | 106 |
| Gemini-2.5 | GPT-4o | Gujarati | 2e-15 | (1.0, 1.52) | 8e-9 | (1.33, 2.41) | 2e-11 | (3.43, 5.1) | 106 |
| Gemini-2.5 | Gemma-3 | Gujarati | 1e-11 | (0.56, 0.98) | 5e-4 | (0.33, 1.33) | 1e-7 | (2.15, 3.95) | 106 |
| Gemini-2.5 | GPT-4o | Telugu | 1e-13 | (0.97, 1.49) | 1e-9 | (1.13, 2.13) | 1e-11 | (3.42, 5.16) | 106 |
| Gemini-2.5 | Gemma-3 | Telugu | 2e-8 | (0.47, 1.07) | 1e-3 | (0.36, 1.3) | 1e-5 | (1.22, 3.03) | 106 |
| Gemini-2.5 | GPT-4o | English | 5e-12 | (0.59, 1.02) | 2e-2 | (0.12, 1.13) | 5e-13 | (4.2, 5.88) | 106 |
| Gemini-2.5 | Gemma-3 | English | 7e-11 | (0.58, 0.94) | 9e-5 | (0.64, 1.85) | 4e-11 | (3.15, 4.79) | 106 |

## C.2 Reliability of LLM-as-a-judge

To remain consistent with prior work Vayani et al. (2025) and to avoid dependence on the Gemini family (which was used earlier for draft generation), we adopt GPT-4o as our LLM-as-a-judge. To evaluate its reliability and potential bias, we conduct a human study on two question types including Short-Answer (Q1) and Adversarial, using the English VQA-Parallel subset (two models; 106 samples), with two independent human annotators.

Across both question types, GPT-4o matches or exceeds human–human agreement (see Table 16). For Short-Answer Q1, human–human Pearson correlations range from $r = 0.66$ to $r = 0.68$, while GPT-4o vs. human-mean correlations range from $r = 0.74$ to $r = 0.79$. For Adversarial questions, human–human agreement ranges from $r = 0.64$ to $r = 0.74$, and GPT-4o correlates with the human-mean at $r = 0.73$ to $r = 0.80$. These results indicate that GPT-4o achieves human-level consistency, supporting its use as a reliable and scalable judge. However, we also acknowledge that LLM-as-a-judge might still reflect some bias where the judge may not fully capture all cultural nuances, introducing a layer of uncertainty.

Table 16: **Correlation results** for Human judges and LLM judge (GPT-4o) for Short-Answer (Q1) and Adversarial questions. We used Pearson correlation to report results.

| Model | Short-Answer (Q1) | | Adversarial | |
|---|---|---|---|---|
| | Human–Human | GPT-4o vs Human-Mean | Human–Human | GPT-4o vs Human-Mean |
| Gemini-2.5 | 0.66 | 0.74 | 0.74 | 0.80 |
| GPT-4o | 0.68 | 0.79 | 0.64 | 0.73 |

## D Dataset Analysis and Benchmark details

We provide more details about our dataset here. Figure 5 shows that the dataset spans diverse cultural categories, with the largest shares in *Heritage (12.4%)*, *Religion (11.2%)*, *Architecture (11.1%)*, *Food (8.6%)*, and *Lifestyle (8.1%)*. The numbers of these categories are provided in Figure 3 with 4698 and 4212 questions for Heritage and Religion respectively. Also, Hindi has the largest share of QA pairs (26.8%) followed by Kannada (11.9%) and Tamil (9.7%) among Indic languages. Interestingly, Adversarial questions have the shortest length (15) while their answers have greater length (44) than Short-QAs (18 and 21). Meanwhile in IVB-MMT, the caption length in number of words follows a distribution with mean $= 131.30$ and std $= 43.52$ for Hindi. We expect that the distribution for other languages will also be the same. On the other hand, in the OCR track, the average number of words for different languages vary significantly with Hindi (329) and Gujarati (247) topping the table and Tamil (121) and Malayalam (127) being the least. Please refer to Table 18 for details of the dataset. Moreover, Table 19 shows the State/UT-wise image distribution of the data set. Figure 13 presents word clouds by category, where the word 'traditional' emerges as a dominant term across domains, underscoring the cultural grounding of the benchmark. Alongside, category-specific concepts appear prominently e.g., sweet and dish in Food, palace and temple in Heritage, dance and instrument in Music, and buddhist and church in Religion. This distribution confirms that IVB-VQA emphasizes India's traditional practices while maintaining diversity across food, heritage, festivals, lifestyle etc. We also included the adversarial questions in our dataset to probe for the ability to reject false cultural assumptions. For example: "What kind of rituals are performed for the Hindu deity Krishna in this temple?" where the image depicts a Jain temple, not a Hindu temple. All adversarial prompts were manually validated to ensure the false presuppositions are realistic, culturally plausible, and not trivially detectable. These questions specifically test challenging multimodal reasoning skills: detecting presupposition errors, handling culturally specific counterfactuals (see Figure 9), and rejecting misleading cues.

Following prior work (Vayani et al., 2025), we use LLM-generated seed questions to accelerate the creation of six QA types per image, while explicitly instructing annotators to substantially revise or rewrite any questions lacking cultural grounding. To validate our choice of the Gemini family for this initial synthesis step, we conducted a small human preference pilot study comparing Gemini-2.5-Flash and GPT-4o across English QA generation and translations (Hindi/Telugu). Gemini provides comparable quality for English QA (48 vs. 34 votes) and clear advantages for Hindi and Telugu

translations (40 vs. 25; 55 vs. 23). Given this quality profile and its substantially lower generation cost (see Table 17), we select Gemini-2.5-Flash as our synthesis model.

Table 17: **Cost comparison for Gemini and OpenAI models in OCR and Captioning tasks (rounded to 2 decimals in $)**. We provide an approximate cost at the time of submission for a sample of 1000 images based on the assumptions of input and outputs tokens. Batch APIs are half the price of Single calls. Here Gemini-2.5-F denotes Gemini-2.5-Flash and Gemini-2.5-P denotes Gemini-2.5-Pro. Our benchmark further involved a multiple for number of languages and questions.

| Task | #Images | Input (M) | Output (M) | Batch (Gemini) | | Single (Gemini) | | Batch (OpenAI) | | Single (OpenAI) | |
|---|---|---|---|---|---|---|---|---|---|---|---|
| | | | | Gemini-2.5-F | Gemini-2.5-P | Gemini-2.5-F | Gemini-2.5-P | GPT-4o | GPT-4o-mini | GPT-4o | GPT-4o-mini |
| OCR | 1,000 | 0.05 | 0.30 | 0.58 | 2.34 | 1.15 | 4.68 | 2.52 | 2.01 | 5.04 | 4.01 |
| Captioning | 1,000 | 0.05 | 0.15 | 0.39 | 1.59 | 0.78 | 3.18 | 1.77 | 1.96 | 3.54 | 3.92 |

We emphasize that IndicVisionBench uses Indian states as a proxy for cultural groups. The language distribution in the benchmark is therefore a direct outcome of our *state-wise, culture-first* sampling strategy, in which each state is mapped to its most widely spoken language. Several of India's largest states like Uttar Pradesh, Madhya Pradesh, Rajasthan, Haryana, Bihar, Uttarakhand, and Delhi are predominantly Hindi-speaking. As a result, when aiming for balanced *cultural* rather than strictly *linguistic* coverage, a higher proportion of Hindi QA pairs naturally emerges. All images were annotated and translated according to the primary language associated with their corresponding state. Our design goal was to reflect India's geographic and cultural diversity rather than enforce an artificial uniform distribution across languages.

Our subset selection follows directly from this state-wise design: each collected image is associated with a specific Indian state or union territory, either through metadata or through culturally identifiable attributes. This mapping guided the creation of the VQA-Indic subset, where translated QA pairs were chosen to maintain region-level balance. Many crowdsourced images already contained explicit cultural or regional cues, making them natural candidates for creating Indic-language QA in the corresponding state language.

**VQA-Indic.** The translated subset was constructed to preserve geographic and cultural diversity across regions. Rather than random or convenience sampling, we selected images to ensure broad state-level representation, resulting in a culturally grounded, geographically balanced translated set.

**VQA-Parallel (106 images).** The VQA-Parallel subset was intentionally designed as a controlled multilingual evaluation slice rather than a large-scale one. We selected 106 images such that: (i) all 10 Indic languages are represented without over-emphasizing any specific region, and (ii) all 13 cultural topics are represented. This produced a consistent and balanced set suitable for high-quality human translation into 11 languages for controlled cross-lingual comparison.

Table 18: **Summary of IndicVisionBench datasets.**

| Task | #Images | Languages | Type |
|---|---|---|---|
| OCR | 876 | 10 | Image–text pairs |
| VQA-EN | 4117 | English | 6 QA types |
| VQA-Indic | 1007 | 10 | Indic langs QA |
| VQA-Parallel | 106 | English+10 | Parallel QA |
| MMT | 106 | English+10 | Parallel captions |

Table 19: **State/UT-wise image distribution in IndicVisionBench-VQA.**

| State/UT | #Images | State/UT | #Images |
|---|---|---|---|
| Andaman & Nicobar | 97 | Madhya Pradesh | 98 |
| Andhra Pradesh | 107 | Maharashtra | 128 |
| Arunachal Pradesh | 99 | Manipur | 100 |
| Assam | 101 | Meghalaya | 75 |
| Bihar | 120 | Mizoram | 78 |
| Chandigarh | 100 | Nagaland | 94 |
| Chhattisgarh | 90 | Odisha | 116 |
| Dadra & Nagar Haveli, Daman & Diu | 54 | Puducherry | 106 |
| Delhi | 141 | Punjab | 108 |
| Goa | 101 | Rajasthan | 131 |
| Gujarat | 110 | Sikkim | 97 |
| Haryana | 99 | Tamil Nadu | 139 |
| Himachal Pradesh | 99 | Telangana | 111 |
| Jammu & Kashmir | 105 | Tripura | 97 |
| Jharkhand | 94 | Uttar Pradesh | 129 |
| Karnataka | 242 | Uttarakhand | 112 |
| Kerala | 116 | West Bengal | 109 |
| Ladakh | 99 | Pan-India | 320 |
| Lakshadweep | 101 | – | – |

## D.1 TOPICS COVERED

> **Topics covered using IndicVisionBench**
>
> The categories that we have for the crawled images are as follows:
>
> - **Food**: Iconic regional cuisines and dishes
> - **Lifestyle**: Traditional attire, daily routines, and modern practices
> - **Literature**: Renowned works, authors, and poets
> - **Music and Dance**: Classical, folk, and traditional performance arts
> - **Religion**: Major faiths, rituals, and festivals
> - **Customs**: Cultural etiquette and greeting practices
> - **Festivals**: National and regional celebrations
> - **Heritage**: Monuments, sites, and landmarks of historical importance
> - **Economy**: Key industries and occupations
> - **Media**: Popular entertainment figures, cinema, and television
> - **Architecture**: Traditional Art and Architecture
> - **Sports**: Indigenous games and traditional sports
> - **Notable Figures**: Influential leaders and historical personalities

Table 20: **Comparison of existing VQA evaluation datasets with IndicVisionBench.** IndicVisionBench supports 3 multi-lingual tasks compared to existing benchmarks.

| Dataset | No. Questions | No. Images | Multilingual? | Task Format | Culturally Diverse Images? |
|---|---|---|---|---|---|
| MaXM (Changpinyo et al., 2023) | 2,142 | 335 | ✓ | VQA | No |
| GDVCR (Yin et al., 2021) | 886 | 328 | ✗ | VQA | Yes |
| MaRVL (Liu et al., 2021) | 5,670 | 4,914 | ✓ | VQA | Yes |
| CVQA (Romero et al., 2024) | 9,044 | 4,560 | ✓ | VQA | Yes |
| CulturalVQA(Romero et al., 2024) | 2,378 | 2,328 | ✗ | VQA | Yes |
| ALM-Bench(Vayani et al., 2025) | 22,763 | 2,328 | ✓ | VQA | Yes |
| IndicVisionBench | **37,740** | **4,993** | ✓ | VQA, OCR, MMT | Yes |

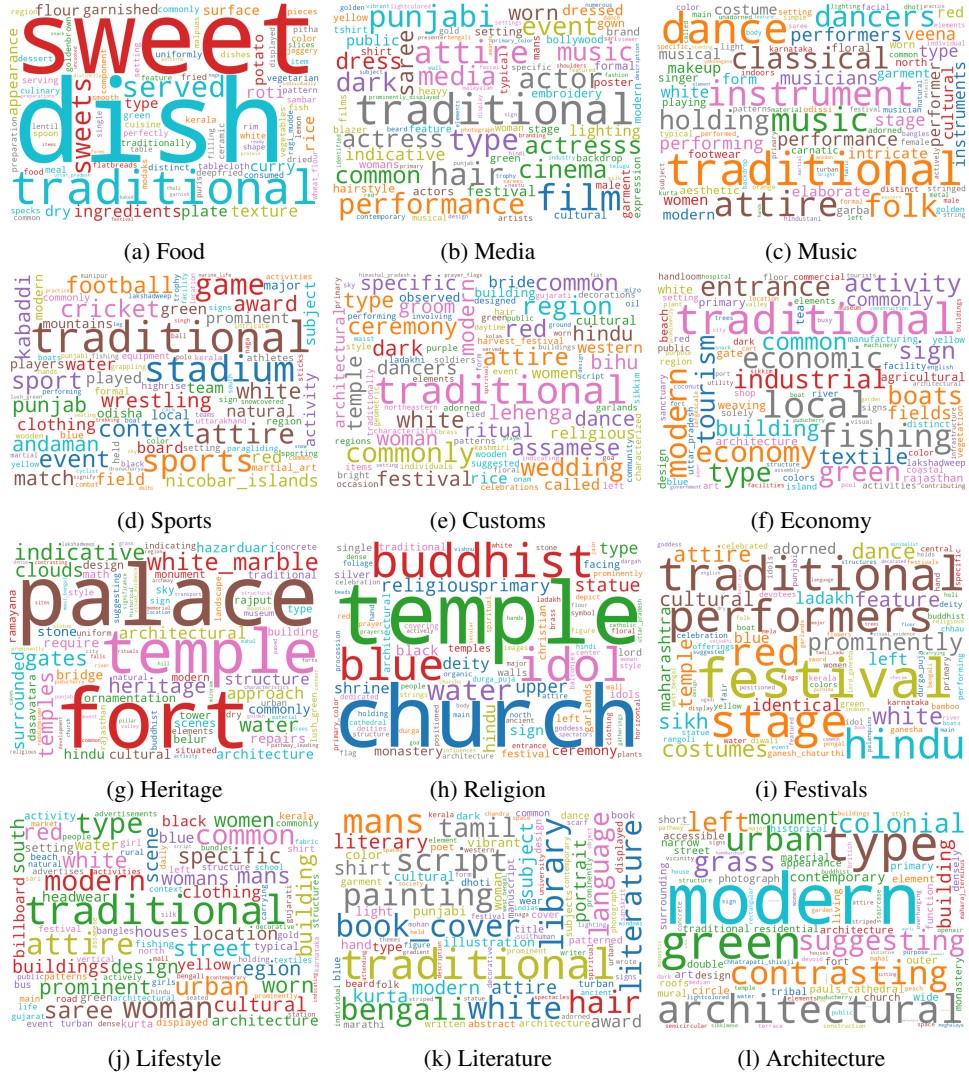

(a) Food

(b) Media

(c) Music

(d) Sports

(e) Customs

(f) Economy

(g) Heritage

(h) Religion

(i) Festivals

(j) Lifestyle

(k) Literature

(l) Architecture

Figure 13: **Word clouds of different categories in IndicVisionBench-VQA.** We omit the words "India" & "Indian" in the word clouds to show other important topics.

Table 21: **Evaluation metrics used for different tasks in IndicVisionBench** highlighting deterministic and non-deterministic measures along with their rationale.

| Task | Deterministic | Non-deterministic | Rationale |
|------|--------------|-------------------|-----------|
| OCR | ANLS, WER, CER | – | Robustness to script |
| VQA | Exact Match | LLM-as-a-Judge | QA accuracy + reasoning quality |
| MMT | BLEU, RIBES | – | Translation quality |

## D.2 EXAMPLES OF OUR DATASET

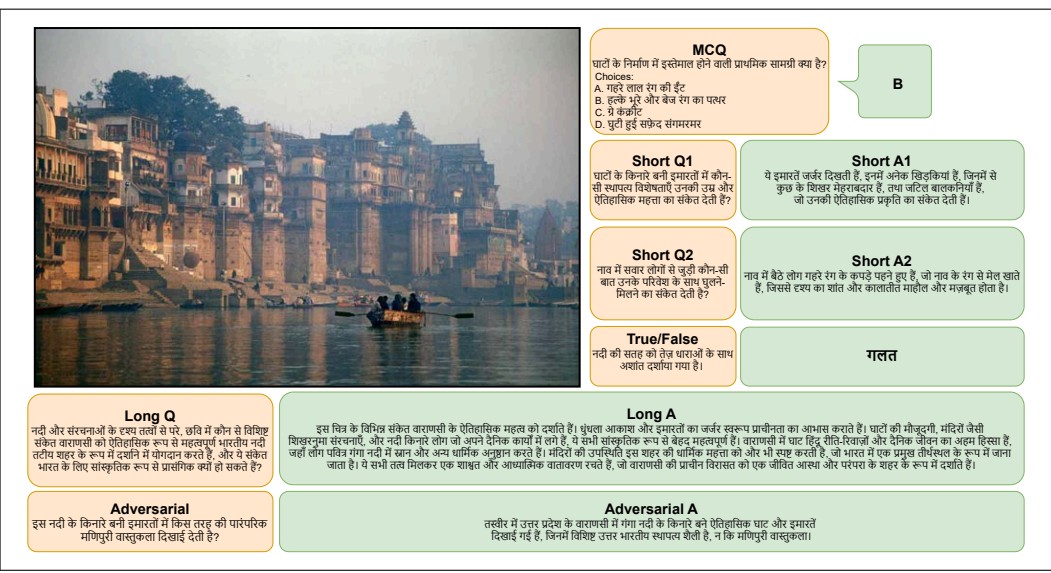

Figure 14: **Example from IndicVisionBench-VQA** for Hindi

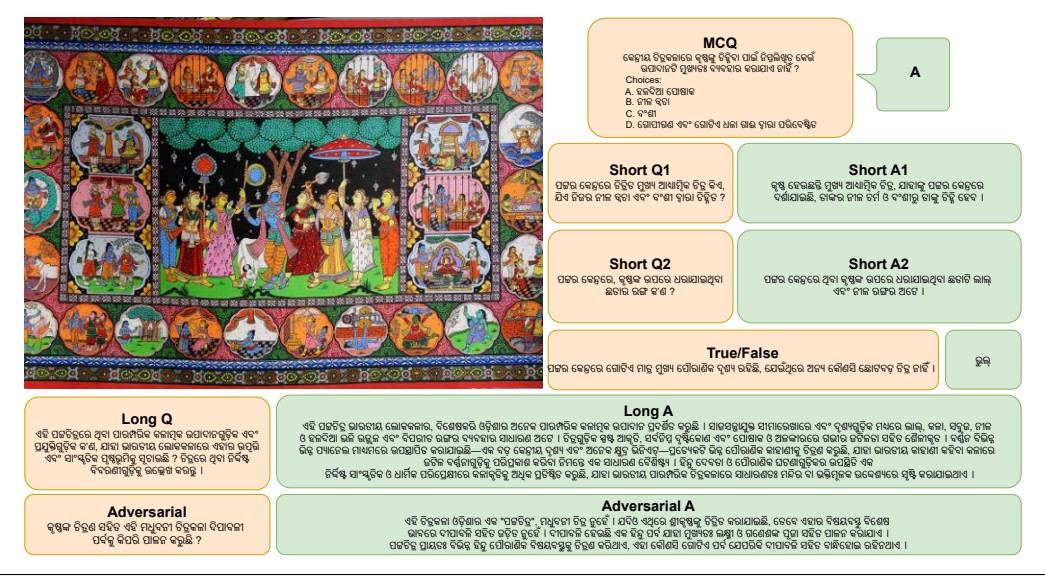

Figure 15: **Example from IndicVisionBench-VQA** for Odia

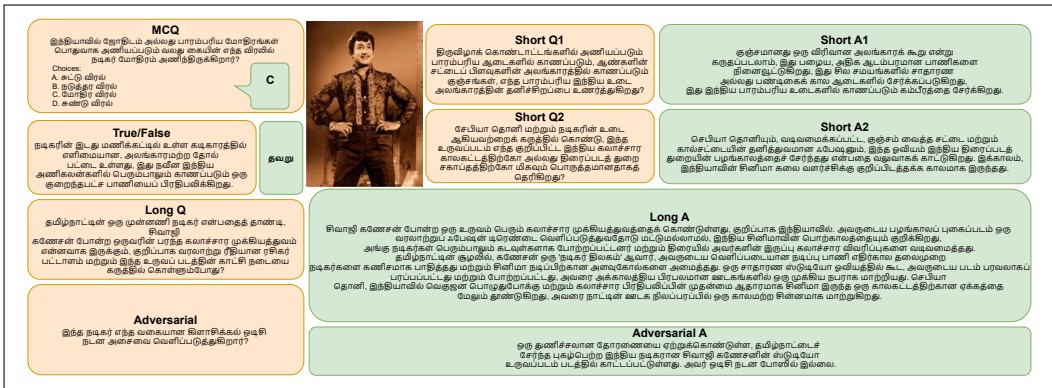

Figure 16: **Example from IndicVisionBench-VQA** for Tamil

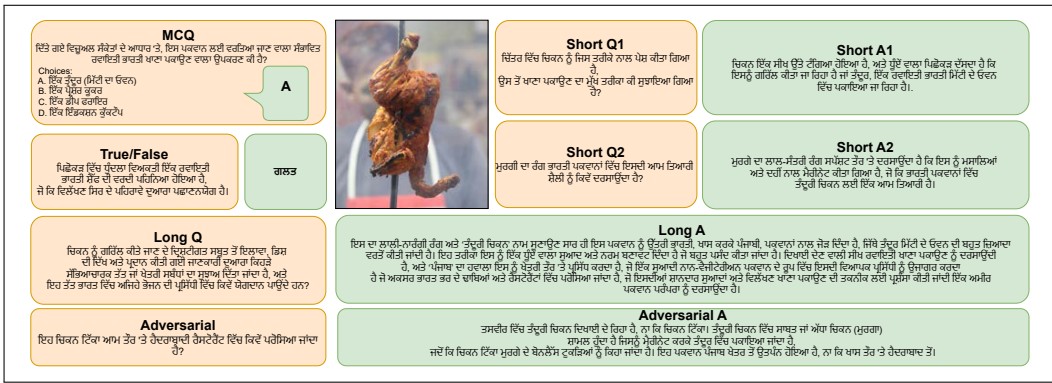

Figure 17: **Example from IndicVisionBench-VQA** for Punjabi

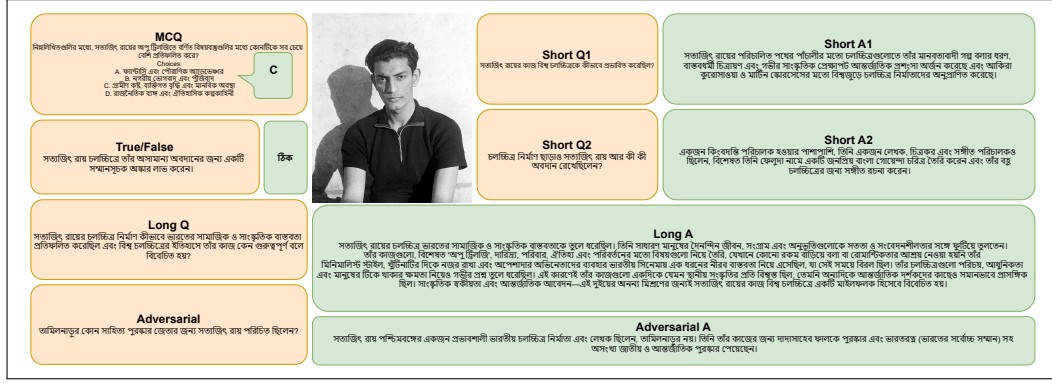

Figure 18: **Example from IndicVisionBench-VQA** for Bengali

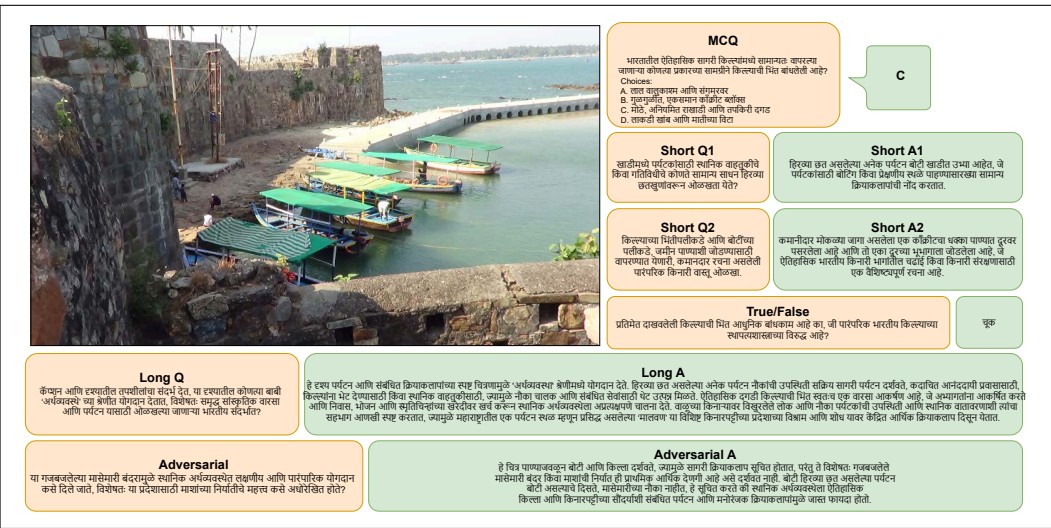

Figure 19: **Example from IndicVisionBench-VQA** for Marathi

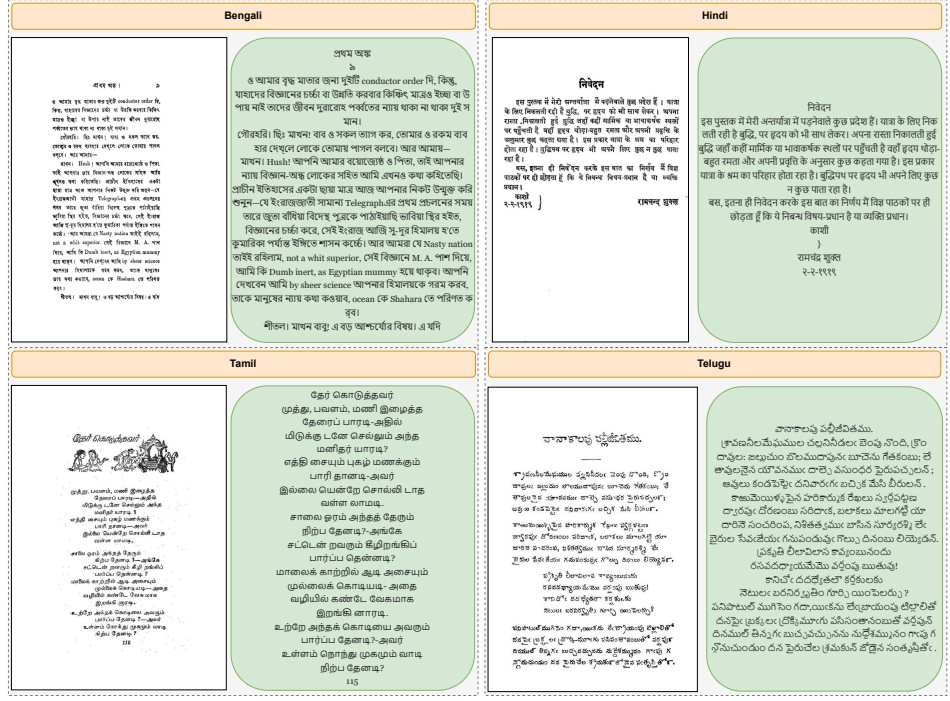

Figure 20: **Examples of IndicVisionBench-OCR.** We show documents and the corresponding Ground Truth texts in Bengali, Hindi, Tamil and Telugu.

| Bengali | English | Malayalam |
|---|---|---|

গোঁফ ও দাড়িওয়ালা একজন পুরুষ কৃষক, একটি কমলা রঙের ফুলহাতা শার্ট, ধূসর প্যান্ট এবং একটি লাল ও সাদা চেকের পাগড়ি পরা অবশ্য একটি বিশাল সোনালী গমের ক্ষেতে পরিষ্কার আকাশের নিচে বসে আছেন। তিনি তার হাতে একটি রূপালী কাস্তে ধরে আছেন, যা তার হাঁটুর উপর রাখা আছে এবং এবং সরাসরি ক্যামেরার দিকে তাকিয়ে আছেন। তার পাশে মাটিতে কাটা গমের আঁটি দেখা যাচ্ছে এবং পটভূমিতে আরও গমের শীষ আছে। ফ্রেমের বাম দিকে অন্য একজন ব্যক্তির লাল শার্ট আংশিকভাবে দৃশ্যমান।

A male farmer, with a mustache and beard, wearing an orange full-sleeved shirt, grey pants, and a red and white checkered turban, squats in a large golden wheat field under a clear sky. He holds a silver sickle in his hands, which are resting on his knees, and looks directly at the camera. Bundles of cut wheat stalks are visible on the ground next to him, with more standing wheat stalks filling the background. Another person's red shirt is partially visible on the left side of the frame.

മീശയും താടിയുമുള്ള ഒരു പുരുഷ കർഷകൻ ഓറഞ്ച് നിറത്തിലുള്ള ഫുൾ സ്ലീവ് ഷർട്ടും ചാരനിറത്തിലുള്ള പാൻറ്റ്സും ചുവപ്പും വെള്ളയും കലർന്ന കള്ളി തലക്കെട്ട് ധരിച്ച് തെളിഞ്ഞ ആകാശത്തിന് കീഴെ സ്വർണ്ണനിറമുള്ള വലിയ ഗോതമ്പ് പാടത്ത് കുത്തിയിരിക്കുന്നു. അയാൾ തന്റെ കാൽമുട്ടിൽ വെച്ചിരിക്കുന്ന കൈകളിൽ ഒരു വെള്ളി അരിവാൾ പിടിച്ച് ക്യാമറയിലേക്ക് നോക്കുന്നു. മുറിച്ച ഗോതമ്പ് കറ്റകൾക്ക് സമീപത്ത് കാണാം. കൂടുതൽ ഗോതമ്പ് കറ്റകൾ പശ്ചാത്തലത്തിൽ നിറഞ്ഞുനിൽക്കുന്നു. ഫ്രെയിമിന്റെ ഇടതുവശത്ത് മറ്റൊരാളുടെ ചുവന്ന ഷർട്ട് ഭാഗികമായി കാണാം.

| Tamil | | Telugu |
|---|---|---|

மீசையும் தாடியும் கொண்ட ஒரு ஆண் விவசாயி, ஆரஞ்சு நிற முழுக்கை சட்டை, சாம்பல் நிற கால்சட்டை மற்றும் சிவப்பு வெள்ளை நிற கட்டம்போட்ட தலைப்பாகை அணிந்து, தெளிவான வானத்தின் கீழ் ஒரு பெரிய பொன்னிற கோதுமை வயலில் குதிகால்கீட்டு அமர்ந்திருக்கிறார். அவர், தனது கைகளில் ஒரு வெள்ளி நிற அரிவாளை வைத்திருக்கிறார். அது அவரது முழங்கால்களின் மேல் வைக்கப்பட்டிருக்கிறது. மேலும், அவர் நேரடியாக கேமராவைப் பார்க்கிறார். வெட்டப்பட்ட கோதுமை தண்டுகளின் கட்டுகள் அவருக்கு அருத்த தரையில் காணப்படுகின்றன, மேலும், பின்னணியில் விளைந்த நிற்கும் கோதுமை பயிர்கள் நிரம்பியுள்ளன. சட்டகத்தின் இடது பக்கத்தில் மற்றொரு நபரின் சிவப்பு சட்டை ஓரளவு தெரிகிறது.

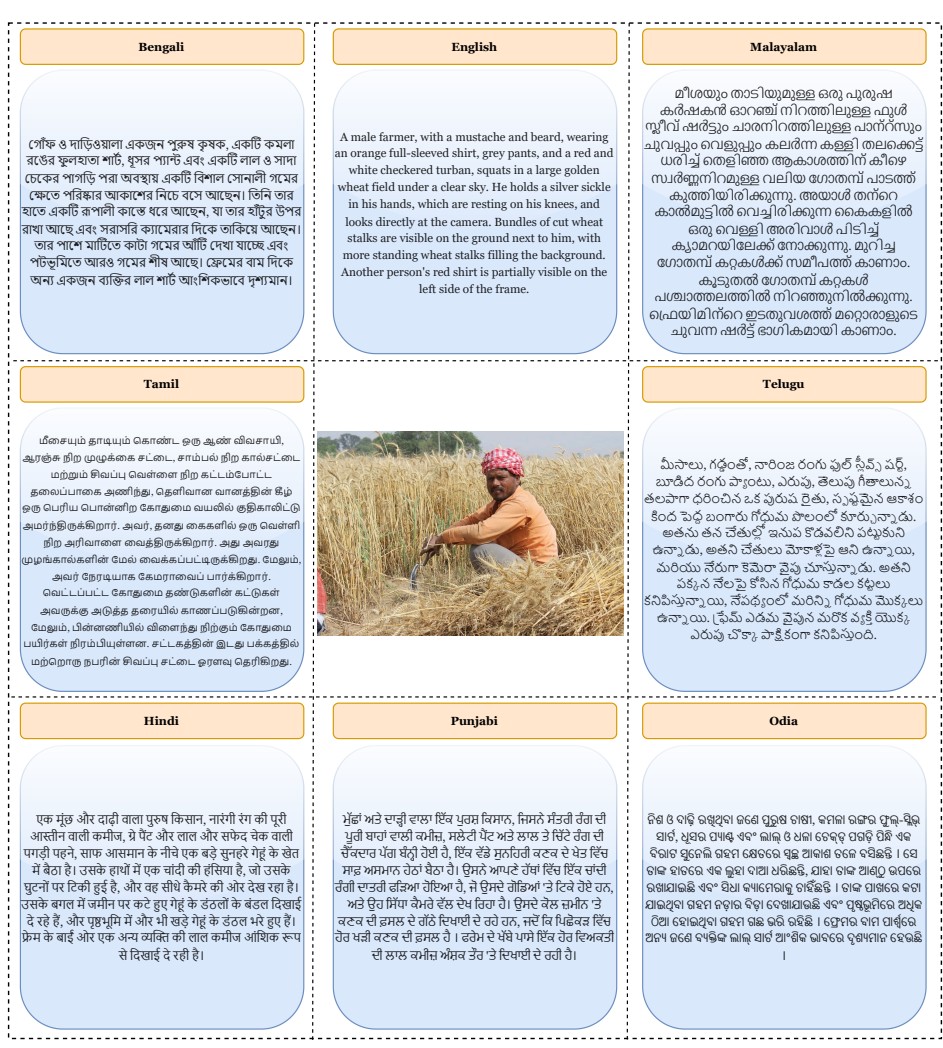

మీసాలు, గడ్డంతో, నారింజ రంగు ఫుల్ స్లీవ్స్ షర్ట్, బూడిద రంగు ప్యాంటు, ఎరుపు, తెలుపు గీతాలున్న తలపాగా ధరించిన ఒక పురుష రైతు, స్పష్టమైన ఆకాశం కింద పెద్ద బంగారు గోధుమ పొలంలో కూర్చున్నాడు. అతను తన చేతుల్లో ఇనుప కొడవలిని పట్టుకుని ఉన్నాడు, అతని చేతులు మోకాళ్లపై ఆని ఉన్నాయి, మరియు నేరుగా కెమెరా వైపు చూస్తున్నాడు. అతని పక్కన నేలపై కోసిన గోధుమ కాడల కట్టలు కనిపిస్తున్నాయి, నేపథ్యంలో మరిన్ని గోధుమ మొక్కలు ఉన్నాయి. ఫ్రేమ్ ఎడమ వైపున మరొక వ్యక్తి యొక్క ఎరుపు చొక్కా పాక్షికంగా కనిపిస్తుంది.

| Hindi | Punjabi | Odia |
|---|---|---|

एक मूंछ और दाढ़ी वाला पुरुष किसान, नारंगी रंग की पूरी आस्तीन वाली कमीज, ग्रे पैंट और लाल और सफेद चेक वाली पगड़ी पहने, साफ आसमान के नीचे एक बड़े सुनहरे गेहूं के खेत में बैठा है। उसके हाथों में एक चांदी की हंसिया है, जो उसके घुटनों पर टिकी हुई है, और वह सीधे कैमरे की ओर देख रहा है। उसके बगल में जमीन पर कटे हुए गेहूं के डंठलों के बंडल दिखाई दे रहे हैं, और पृष्ठभूमि में और भी खड़े गेहूं के डंठल भरे हुए हैं। फ्रेम के बाईं ओर एक अन्य व्यक्ति की लाल कमीज आंशिक रूप से दिखाई दे रही है।

ਮੁੱਛਾਂ ਅਤੇ ਦਾੜ੍ਹੀ ਵਾਲਾ ਇੱਕ ਪੁਰਸ਼ ਕਿਸਾਨ, ਜਿਸਨੇ ਸੰਤਰੀ ਰੰਗ ਦੀ ਪੂਰੀ ਬਾਹਾਂ ਵਾਲੀ ਕਮੀਜ਼, ਸਲੇਟੀ ਪੈਂਟ ਅਤੇ ਲਾਲ ਤੇ ਚਿੱਟੇ ਰੰਗ ਦੀ ਚੈੱਕਦਾਰ ਪੱਗ ਬੰਨ੍ਹੀ ਹੋਈ ਹੈ, ਇੱਕ ਵੱਡੇ ਸੁਨਹਿਰੀ ਕਣਕ ਦੇ ਖੇਤ ਵਿੱਚ ਸਾਫ਼ ਅਸਮਾਨ ਹੇਠਾਂ ਬੈਠਾ ਹੈ। ਉਸਨੇ ਆਪਣੇ ਹੱਥਾਂ ਵਿੱਚ ਇੱਕ ਚਾਂਦੀ ਰੰਗੀ ਦਾਤਰੀ ਫੜੀਆਂ ਹੋਈਆਂ ਹੈ, ਜੋ ਉਸਦੇ ਗੋਡਿਆਂ 'ਤੇ ਟਿਕੇ ਹੋਏ ਹਨ, ਅਤੇ ਉਹ ਸਿੱਧਾ ਕੈਮਰੇ ਵੱਲ ਦੇਖ ਰਿਹਾ ਹੈ। ਉਸਦੇ ਕੋਲ ਜ਼ਮੀਨ 'ਤੇ ਕਟਕ ਦੀ ਫਸਲ ਦੇ ਗੱਠੇ ਦਿਖਾਈ ਦੇ ਰਹੇ ਹਨ, ਜਦੋਂ ਕਿ ਪਿੱਛੇਕਰ ਵਿੱਚ ਹੋਰ ਖੜ੍ਹੀ ਕਣਕ ਦੀ ਫਸਲ ਹੈ। ਫਰੇਮ ਦੇ ਖੱਬੇ ਪਾਸੇ ਇੱਕ ਹੋਰ ਵਿਅਕਤੀ ਦੀ ਲਾਲ ਕਮੀਜ਼ ਅੰਸ਼ਕ ਤੌਰ 'ਤੇ ਦਿਖਾਈ ਦੇ ਰਹੀ ਹੈ।

ଦିଶ ଓ ଦାଢ଼ି ରଖିଥିବା ଜଣେ ପୁରୁଷ ଚାଷୀ, କମଲା ରଙ୍ଗର ଫୁଲ୍-ସ୍ଲିଭ୍ ସାର୍ଟ, ଧୂସର ପ୍ୟାଣ୍ଟ ଏବଂ ଲାଲ୍ ଓ ଧଳା ଚେକ୍ସ ପ୍ରଥମ ଟୋପି ... ଏକ ବିଶାଳ ସୁନେଲି ଗହମ ଚ୍ଷେତରେ ସ୍ୱଛ ଆକାଶ ତଳେ ବସିଛନ୍ତି । ସେ ତାଙ୍କ ହାତରେ ... ଲୁହା ଦାଆ ଧରିଛନ୍ତି, ଯାହା ... ରଖାଯାଇଛି ଏବଂ ... ... କେମେରାକୁ ଚାହିଁଛନ୍ତି । ତାଙ୍କ ପାଖରେ ଭୂଇଁରେ କଟା ... ... ରଖାଯାଇଛି ଏବଂ ... ... ଗହମଗଛ ଭର୍ତ୍ତି ... ... ଫ୍ରେମର ବାମ ... ... ଜଣେ ବ୍ୟକ୍ତିର ଲାଲ୍ ସାର୍ଟ ... ... ଦେଖାଯାଉ ... ।

Figure 21: **IndicVisionBench-MMT benchmark.** We show an example image and corresponding translations of the caption in 8 languages.

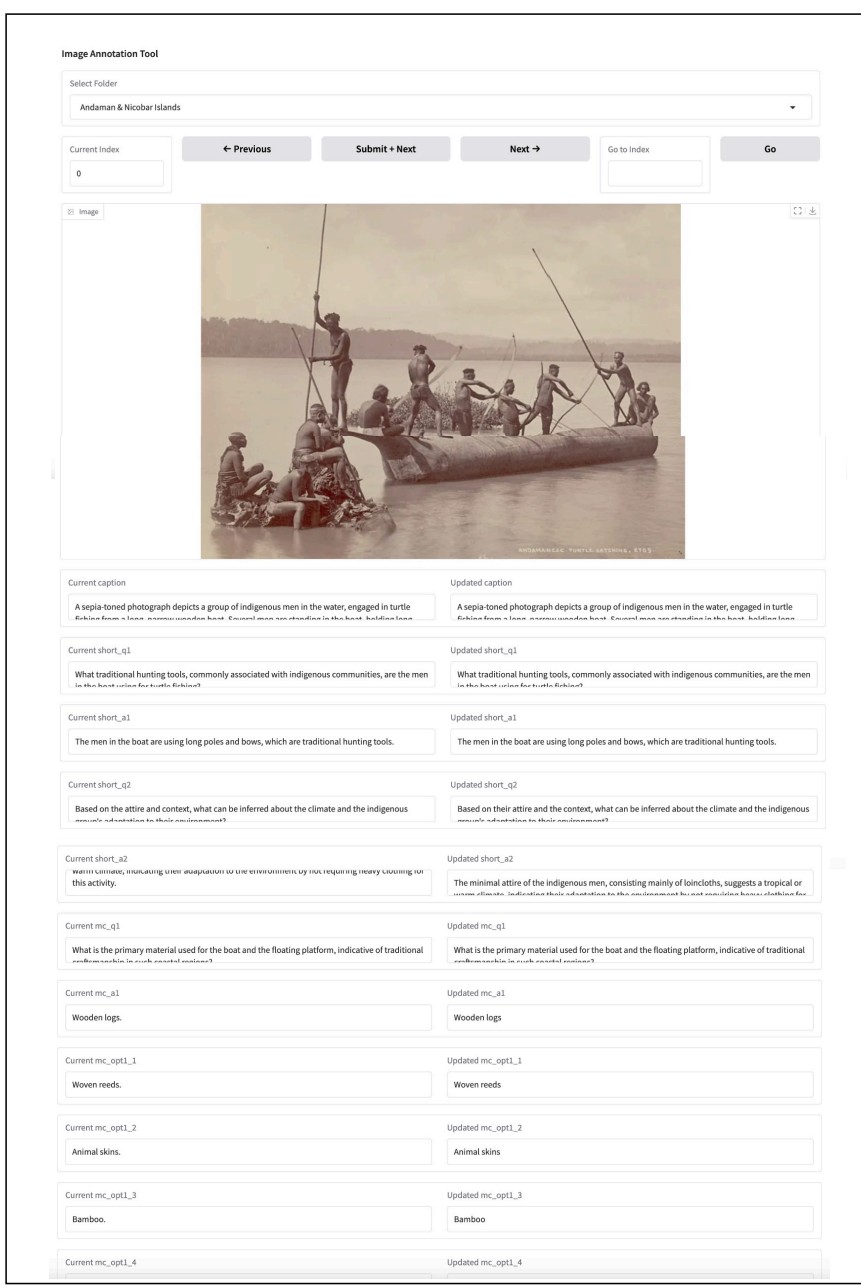

Figure 22: **Image QA pairs' correction tool.** Interface of the QA pairs' correction tool provided to the human annotators.

# E    HUMAN ANNOTATIONS

To quantify the extent of human correction in our benchmark, we compute the Edit Rate and Average Relative Edit, defined as:

$$\text{Edit Rate} = \frac{E}{N} \tag{1}$$

$$\text{Avg. Relative Edit} = \frac{1}{E} \sum_{i=1}^{E} \frac{\text{Levenshtein Distance (Edited instance, Original instance)}}{\text{Length (Original instance)}} \tag{2}$$

where, $N$ is the total number of instances and $E$ is the number of edited instances.

Across all stages of dataset construction, human annotators applied extensive revisions to the LLM-generated drafts. Table 22 reports the edit rates and average relative edit scores across the three corpora. VQA-En (English QAs of IVB-VQA) exhibits an edit rate of 18.61% and a high average relative edit of 77.2%, indicating that the edits themselves were often substantial. VQA-Indic shows a markedly higher proportion of edited items (62.5%), reflecting its linguistic diversity and greater variability in the initial drafts, with particularly large edits for MCQs (86.1%) and Short-QAs (58.1%). The MMT corpus exhibits an edit rate of 79.1% and an average relative edit of 21.6% per caption, underscoring the extensive human refinement involved in the translation process. A relative edit value of 128% for Levenshtein distance shows that the Short-QA corrections were substantial, often exceeding the length of the original text and reflects major rewrites rather than minor adjustments.

Table 22: Corpus-level human annotation statistics: overall edit rates, overall average relative edit, and per-question-type average relative edit.

| Corpus | Edit Rate (Overall) | Avg. Rel. Edit (Overall) | Avg. Relative Edit (per QA type) | | | | |
|---|---|---|---|---|---|---|---|
| | | | Short-QA | Long-QA | Adv-QA | MCQ | T/F |
| **VQA-EN** | 18.6% | 77.2% | 128.9% | 66.5% | 54.4% | 53.5% | 31.1% |
| **VQA-Indic** | 62.5% | 48.9% | 58.1% | 37.2% | 22.9% | 86.1% | 25.2% |

Across languages, Hindi shows the highest edit rate (Edit rate 73.7%, 33.2% Avg. relative edit), followed by Kannada (69.7%,30.9%), Bengali (57.6%, 37.2%), and Punjabi (55.0%, 29.8%). Notably, Bengali yields the highest average relative edit, while Hindi has the largest edit rate, with the remaining languages showing moderately high but consistent correction levels. All these statistics highlight the extent of human intervention and confirm that the final benchmark content differs substantially from the initial LLM-generated drafts.

We also report the demographics of our annotation team in Figure 23. For the VQA subset, crowd-sourced images were collected from 18 contributors, including some of the authors, who provided photographs spanning diverse cultural contexts. This group comprised 13 male and 5 female participants (ages 20–50). The QA correction and translation phase was carried out by a team of 25 annotators responsible for refining, validating, and translating all QAs across languages. This team included 13 female and 12 male annotators (ages 20–40), with 8 annotators for English, 3 for Hindi, and 1–2 annotators for each of the remaining Indic languages represented in the benchmark.

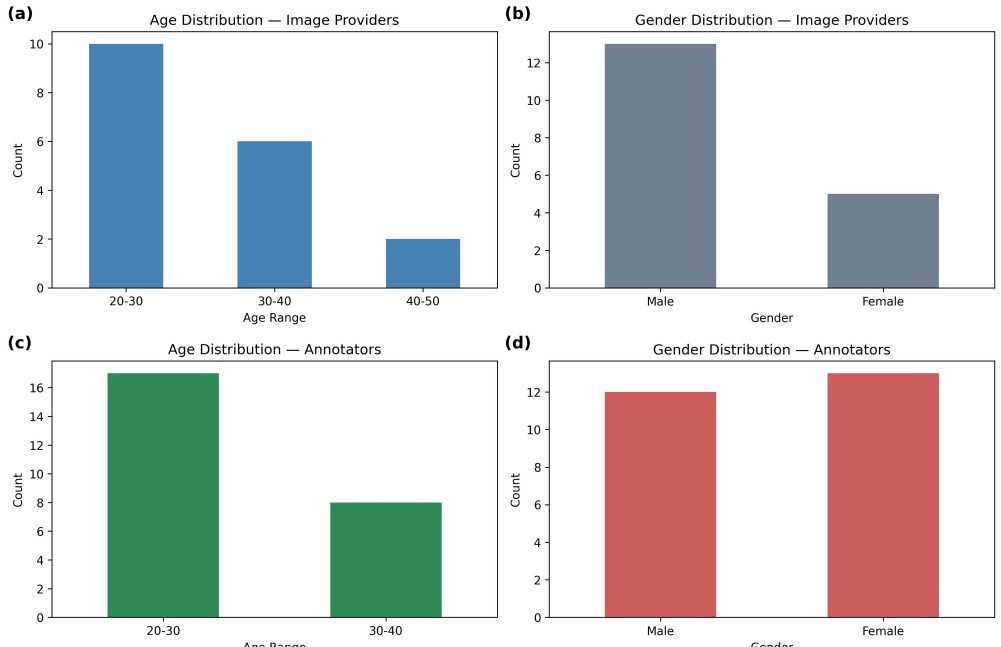

Figure 23: **Demographic distribution.** We shows the demographics of the crowd-sourced image contributors and QA annotators: (a) Age distribution of image providers, (b) Gender distribution of image providers, (c) Age distribution of annotators, and (d) Gender distribution of annotators.

**Annotation guidelines for the IndicVisionBench Cultural QA Dataset**

The task involves enriching image–caption pairs with culturally grounded question–answer (QA) annotations. The focus is on determining whether an image depicts Indian culture and, if so, generating diverse QA pairs that capture cultural specificity.

**Scope**
Annotators assess cultural relevance (e.g., attire, festivals, rituals, food, architecture) and generate QA pairs to support training and evaluation of multimodal models.

**Captions**
Each image is provided with human-annotated keywords and an auto-generated caption. Keywords are fixed; captions may be edited for accuracy. Both should be used as context when forming QA pairs.

**Annotation Process**
- **If an image does not depict Indian culture:**
    - Set depicts_indian_culture = False
    - Set state_specific_culture = no
    - Leave other fields unchanged.
- **If an image does depict Indian culture:**
    - Update caption if necessary.
    - Provide the following QA annotations:
        * **Short Answer QAs:** Two concise cultural QA pairs.
        * **MCQ:** One multiple-choice question with one correct option and three plausible distractors (exact string match required between answer and option).
        * **True/False:** One fact-based cultural question with answer True or False.
        * **Long Answer:** One descriptive QA pair (4–6 lines).

**State-Specific Culture**
If the image reflects a specific Indian state, record the state name; otherwise enter `no`.

**Adversarial Question**
Each culturally relevant image requires one adversarial question designed to include a plausible but incorrect cultural assumption.
- **Dos:** Specific, confident, culturally relevant, and misleading but realistic.
- **Don'ts:** Speculative ("Is this...?" ), vague, hedged, or trivial.
- **Examples:** Mistaking Aipan art for Bikaneri art, Pongal for Eid, etc.

**Bad or Irrelevant Images**
Low-quality, generic, or culturally irrelevant images are marked as:
- depicts_indian_culture = False
- state_specific_culture = no

**Checklist**
- Verify cultural relevance.
- Mark depicts_indian_culture and state_specific_culture appropriately.
- If relevant, complete all QA fields.
- Do not leave fields empty.

**Note.** Annotations must remain brief, factual, and faithful to Indian culture. Careful, consistent labeling ensures the dataset's reliability for benchmarking multimodal models.

**IndicVisionBench Image Collection Guidelines**

The goal is to curate culturally relevant, open-license images across Indian states for the IndicVisionBench vision-language benchmark. Each state–category pair must contain at least 10 high-quality images that satisfy the following requirements:

1. **Cultural Authenticity**
   Images should accurately reflect Indian cultural elements tied to the specified state.

2. **Category Relevance**
   Each image must belong to one of the predefined categories (e.g., food, literature, festivals).

3. **Geographic Specificity**
   Content should be clearly associated with a specific Indian state.

4. **Open Licensing**
   Only images under Creative Commons licenses permitting commercial use are eligible.

**Collection Process**
1. Formulate search queries in the format: "<category> in <state>" (e.g., Music in Nagaland, Traditional sports in Kerala).
2. Apply Creative Commons usage rights filters.
3. Manually inspect results for authenticity and verify license details.
4. Select as many images per category–state pair as possible, organize them into subfolders, and record metadata (filename, source URL, category, license).

**Exclusions**
- Low-quality, blurry, watermarked, or stereotypical content.
- Images not clearly tied to culture or state.
- Content with unclear or invalid licensing.

**Target Categories**
Food, Lifestyle, Literature, Music & Dance, Religion, Customs, Festivals, Heritage, Economy, Media & Entertainment, Architecture, Sports, Notable Figures.

**Submission**
Each state folder is expected to contain at least 100 images across categories, with a metadata file. For states with limited available material, collect as many culturally relevant images as possible. All submissions are uploaded to the shared repository.

## F    PROMPTS USED

---
**Instructional Prompts for Each Question Category**

**Long Question:**
{Question} Answer the question in detail in {target_language} language.

**Short Question:**
{Question} Please provide brief, clear responses in {target_language} language.

**Adversarial:**
{Question} Answer the question in detail in {target_language}language.

**Multiple Choice (MCQ):**
Strict Instruction: Respond with only one choice in the format `<A>`, ``, `<C>`, or `<D>`.
Do not include any explanation, reasoning, or extra text.
{Question} Your question here
Choices:
A. Option 1
B. Option 2
C. Option 3
D. Option 4

**True/False:**
Strict Instruction: Respond with only {lang_true} or {lang_false}.
{Question} Your question here
Choices: {lang_true} or {lang_false}

**OCR:**
Extract the exact text from this image using OCR. Respond with only the text.

**Multimodal Translation:**
{Question} Answer the question in detail in {target_language} language.

---

We release all prompts used in our study. These include one prompt for generating four QA types (Long, Short, MCQ, and True/False), and a separate dedicated prompt for adversarial QAs. We design adversarial prompts independently because these questions are more challenging and require detailed instructions. We also include prompts used for evaluation via the LLM-as-a-judge framework, where responses are scored on a 0–10 scale. Furthermore, we provide the prompts we use for each type of question during response generation from different models being evaluated.

---
**Prompt for judging short answer questions via LLM-as-a-judge:**

You are a judge evaluating how well a Vision-Language Model answers short-answer type questions.
Evaluate the model's response based on accuracy and correctness with respect to the Ground Truth answer. Assign a high score when the model's response matches closely with the ground truth and a low score when the response lacks knowledge or is unrelated to the ground truth.

Question: {question}
Ground Truth: {ground_truth}
Model response: {predicted_answer}

Provide a single overall score from 0 to 10 based on the given criteria. Strictly return only the numeric score, without any additional commentary.

---

**Category classification prompt for crowdsourced images:**

You are a cultural content classifier. Given the image and the caption, classify it into one or more of the following **top-level categories**. Use the definitions to guide your classification, but **only return one category name** (e.g., "Food", "Lifestyle") that the image can be best classified into in your answer — no subcategories or descriptions.
- Food: Iconic regional cuisines and dishes
- Lifestyle: Traditional attire, daily routines, and modern practices
- Literature: Renowned works, authors, and poets
- Music and Dance: Classical, folk, and traditional performance arts
- Religion: Major faiths, rituals, and festivals
- Customs: Cultural etiquette and greeting practices
- Festivals: National and regional celebrations
- Heritage: Monuments, sites, and landmarks of historical importance
- Economy: Key industries and occupations
- Media: Popular entertainment figures, cinema, and television
- Architecture: Traditional Art and Architecture
- Sports: Indigenous games and traditional sports
- Notable Figures: Influential leaders and historical personalities

Take help from the detailed caption of this image in this task of categorization. Caption: {caption}
**Respond only with the name of the most relevant category from the list above i.e., Food, Lifestyle, Literature, Music, Religion, Customs, Festivals, Heritage, Economy, Media, Architecture, Sports, Notable Figures**. **Do not give any other response and do not provide any explanation or any unnecessary text**.

**QA pairs creation prompt:**

Here is an India-specific image and the image filename, caption, and category of the image I have on hand.
The image filename is this: {image_filename}
The caption is this: {caption}
The category is this: {category}

I'd like you to generate two short questions and answers, one multiple-choice question and answer, one true/false question and answer, and one long question and answer. Refer to the image filename, category, and caption for context and hints. Take into account the cultural diversity of the category that this image falls under with respect to India.

Follow these rules while designing questions and answers:
1. The question must be answerable only by looking at the image.
2. Ensure that the questions are culturally relevant to India and specific to the image.
3. Make the questions in such a way that someone who is not well aware of Indian culture will find them difficult to answer.
4. Provide answers that are concise, accurate, and directly related to the question.
5. For MCQs, provide 1 correct option and 3 incorrect (but relevant) distractors.
6. For MCQs, the question must be answerable even without the choices.
Example of an invalid question: "What song is not performed by this musician" – not answerable if you don't know the choices.
7. Write all questions fluently in English.
8. Be mindful of cultural sensitivities and avoid stereotyping or misrepresentation.
9. Ensure variety: include identity questions ("What is this?", "Where is this?"), reasoning, referencing, and commonsense knowledge.
10. Generate only in English.
11. For short-answer questions, keep answers brief (1–2 sentences).
12. Make all questions distinct and unique.

Give the answers in the following JSON format and output only valid JSON:

{{
    "short_questions": [ { "question": <question>, "answer": <answer>} ],
    "multiple_choice_questions": [ { "question": <question>, "answer": <answer>, "options":
[<option1>, <option2>, <option3>, <option4>] } ],
    "true_false_question": { "question": <question>, "answer": <answer>},
    "long_question": { "question": <question>, "answer": <answer>}
}}

**Adversarial QA pairs creation prompt:**

You are given an image from India along with its caption and the category it belongs to. Your task is to create an adversarial question for the image — one that makes a confident but subtly incorrect cultural assumption about what is shown, especially with respect to India's diverse regional traditions, foods, festivals, clothing, rituals, and art forms.

Here is the image's caption: {caption}
The category it belongs to is: {category}

Generate:
Adversarial Question: A misleading or culturally incorrect question that confidently makes a specific, wrong assumption which is plausibly close to the truth (e.g., mixing up similar Indian art styles, dishes, festivals, or traditions). The question must not reveal that it is making an assumption — just ask the question normally, as if the incorrect assumption is true.

Avoid yes/no or speculative questions.

Examples:
– How was this Bikaneri folk painting prepared on canvas? (image shows Kumaoni Aipan)
– What are the main ingredients in this chicken tandoori dish? (image shows paneer tikka)
– Which Sikh gurdwara is being shown here? (image shows a Hindu temple)
– How is this Eid offering typically presented in Tamil Nadu? (image shows a Pongal celebration)

Non-examples:
– Is this a Bikaneri art piece?
– Are these Modaks made of coconut?
– Considering this is a South Indian Onam celebration...

The adversarial question should be close enough to the actual content that it forces a model to distinguish finely between culturally similar options and reject the incorrect assumption. You also need to generate the answer for the question that you generate.

Do not include introductions or explanations. Output only the following two fields in exactly this format:
Adversarial Question: <your question here>
Answer: <your answer here>

**Prompt for judging adversarial questions via LLM-as-a-judge:**

You are a judge evaluating how well a Vision-Language Model (VLM) answers adversarial questions. In adversarial questions, the question contains a false assumption about the image.

You are given:
– Question: {question}
– Correct Answer (Gold reference): {ground_truth}
– VLM Answer: {predicted_answer}

Your task:
Evaluate the VLM's answer according to this rubric:

1. **Identifying the false assumption (0 or 5 points):**
   - Award 5 points if the VLM explicitly identifies that the question contains a wrong assumption OR expresses uncertainty/difficulty in confirming the assumption.
   - Award 0 points if the VLM ignores the wrong assumption and answers as if the question were correct.
2. **Identifying what the image is actually about (0–5 points):**
   - Award 0–5 points depending on how well the VLM correctly identifies the real content of the image.
        0 = completely wrong or no attempt.
        1–2 = vague or partially correct.
        3–4 = mostly correct but incomplete.
        5 = fully correct identification.

**Final Score = Assumption Score (0 or 5) + Identification Score (0–5) → 0 to 10.**

Instructions:
– Only output the final score as a number between 0 and 10.
– Do not explain reasoning or repeat answers.
– Always respond with the final score. Do not return a blank response.
– Be fair but consistent: partial credit is encouraged for partial identification.

Now, provide the score.

