# OpenReview forum: "IndicVisionBench: Benchmarking Cultural and Multilingual Understanding in VLMs"
_ICLR.cc/2026/Conference — ICLR 2026 Poster_

### Official Review · Reviewer_w7Rf · 2025-10-30

**Soundness:** 3
**Presentation:** 3
**Contribution:** 3
**Rating:** 8
**Confidence:** 4

**Summary:**

The paper introduces IndicVisionBench which is a culturally grounded multimodal benchmark focused on Indian languages. The benchmark covers three tasks-- VQA, multimodal MT, and OCR. It has ~5k images and 37k+ QA pairs in English and 10 Indic languages. The questions include six types including MCQs, True/False, Short Answers (two versions), long answers and adversarial questions with opposite assumptions. Several multimodal LLMs have been evaluated against the developed benchmark and a comparison has been provided.

**Strengths:**

- The paper addresses three tasks by presenting a benchmark for 10 low-resource Indic languages. Such benchmarks are crucial to evaluate capabilities of multimodal LLMs on less represented languages.
- The benchmark covers the annotation of 13 cultural topics and three tasks--VQA, multimodal MT and OCR. Question have further six categories showing a comparative aspect and abilities of LLMs.
- Eight LLMs have been evaluated covering open and closed source models that provides an inside of performance compared to different Indic cultures and languages.
- Parallel slice for controlled analysis, the 106-image VQA-Parallel + MMT subset is well designed for cross-lingual/model-vs-no-image comparisons.
- Adding adversarial questions with false assumptions provides realistic stress test and it really exposes even strong models.

**Weaknesses:**

- Synthetic data creation: many QAs are first generated by the LLM and then corrected by human. What was the error rate and it should be quantified.
- Many images are Commons Google-search cultural images which are highly likely to have been seen by pretraining corpora.

**Questions:**

As most of the images in the benchmark were collected from Google search under creative common license -- is there any correlation with collecting images from Google search and achieving highest performance from Gemini-2.5 (from Google's DeepMind)?

---

> ### Author Response · Authors · 2025-11-21
>
> We sincerely thank the reviewer for their positive and encouraging feedback. We are grateful that the reviewer considers the benchmark well-motivated and an important resource for evaluating multimodal LLMs in low-resource Indic contexts.
>
> We also appreciate the reviewers' concerns and are glad to address them.
>
> > **W1:** Error rates for Synthetic data creation
>
> We thank the reviewer for highlighting the need to report edit/error statistics. We thus calculated both edit rates and average relative edit across all corpora. Edit rate is the fraction of instances that required human editing while average relative edit quantifies the extent of changes. For each edited instance, we compute the Levenshtein distance between the edited and original text, normalize it by the original text length, and average this value across all edited instances.
>
> Across the benchmark, human annotators made substantial revisions: English questions of VQA show an 18.61% edit rate (avg. relative edit 77.2%), VQA-Indic 62.50% (48.9%), and MMT translations 79.06% (21.62%). These numbers indicate that human reviewers intervened heavily and that the final content is well refined beyond raw LLM drafts. We have added these statistics as well as the annotator demographics in Appendix E of the revised manuscript for transparency.
>
>
> > **W2, Q1:** Concerns about possible data contamination with pre-training corpora
>
> We thank the reviewer for raising this important concern. While we acknowledge some overlap with public web data is unavoidable (as is the case for most vision datasets), we provide two pieces of evidence that support the validity of our results:
>
> - OCR Track (Wikisource scans): IVB-OCR consists entirely of digitized manuscript/page scans sourced from Wikisource. These images are visually distinct from common web imagery and are unlikely to appear in pretraining corpora.
>
> - Crowdsourced images in IVB-VQA: IVB-VQA contains both web-scraped and 615 unique crowdsourced images. These human-contributed images are not indexed online and therefore do not appear in any public pre-training data. We included Table 6 in the Appendix which shows model performance on this crowdsourced subset, where the trend closely matches that of the full VQA dataset.
>
> These findings demonstrate that our results remain consistent even on subsets that are unseen, suggesting that high performance  cannot be solely attributed to potential data overlap. We appreciate the reviewer’s suggestion, which led to this analysis.

---

> > ### Author Response · Authors · 2025-11-27
> >
> > Dear reviewer,
> >
> > Thank you again for your thoughtful and encouraging review. Your feedback helped strengthen the paper. We hope the updates comprehensively address all of your comments.

---

### Official Review · Reviewer_iVmL · 2025-10-31

**Soundness:** 1
**Presentation:** 2
**Contribution:** 2
**Rating:** 2
**Confidence:** 4

**Summary:**

This work presents IndicVisionBench, a large-scale benchmark designed to evaluate Vision-Language Models (VLMs) on culturally grounded and multilingual tasks focused on the Indian subcontinent. The benchmark comprises 5,000 unique images and over 37,000 question-answer pairs spanning 13 cultural topics across English and 10 Indian languages. The evaluation framework encompasses three primary tasks: Visual Question Answering (VQA) with six question types, Optical Character Recognition (OCR), and Multimodal Machine Translation (MMT). The authors evaluate models, including both proprietary systems (Gemini-2.5, GPT-4o) and open-source variants, revealing substantial performance gaps particularly for low-resource languages and culturally specific content.​

**Strengths:**

- **Comprehensive Multi-task Framework**: The benchmark's tri-modal evaluation approach (VQA, OCR, MMT) provides a holistic assessment of VLM capabilities beyond simple question-answering. The inclusion of adversarial questions represents a good approach to probing deeper cultural knowledge beyond surface-level recognition.

-  **Linguistic Coverage** : The benchmark covers 10 Indic languages with diverse scripts.

**Weaknesses:**

- **Limited Methodological Novelty**:
While the cultural focus is valuable, the benchmark construction methodology largely follows established paradigms without introducing novel evaluation frameworks. The reliance on synthetic question generation using commercial models (Gemini-1.5-Flash, Gemini-2.5-Flash) raises concerns about potential biases inherited from these models.

- **Self-Preferential Bias and Evaluation Circularity**:
The most severe methodological flaw is the systematic use of Gemini models throughout the benchmark construction pipeline, followed by its evaluation on the same benchmark. Gemini-1.5-Flash and Gemini-2.5-Flash generate synthetic captions and QA pairs​ and translations across languages​. This creates a circular evaluation where Gemini is assessed on data it helped generate, violating fundamental evaluation principles established in contamination literature. The consistently superior performance of Gemini-2.5 across all tasks becomes suspect given this methodological flaw.​

- **Absence of Design Choice Validation**:
The paper provides no systematic benchmarking for the choice of Gemini variants. While mentioning a "small pilot study and cost considerations," no details, quality assessments, or comparative analysis with alternative generation models are provided. Table 6 shows only cost comparisons without quality metrics.  The benchmark's coverage appears skewed toward certain linguistic groups (Hindi at 26.8% of QA pairs) without adequate justification for this distribution.

 - **Complete Lack of Quality Control Statistics**:
Caption Quality: No validation of synthetic caption accuracy against image content using multimodal embeddings or human verification statistics​.
Human Refinement Rates: The claim that "Human reviewers refined all outputs for factual accuracy and cultural alignment" provides zero statistics on edit rates, acceptance rates, or extent of modifications required​.
Inter-Annotator Agreement: Despite being standard practice for evaluation benchmarks, no IAA scores or methodology details are provided.

- **Inadequately Documented Subset Selection**:
VQA-Indic: The selection of which subset from "4K+ images" was translated lacks stratification methodology, selection criteria, or coverage statistics​
VQA-Parallel: The choice of 106 images for multilingual translation provides no justification for representativeness or selection rationale​
Statistical Significance: With Hindi dominating 26.8% of QA pairs and other languages having minimal representation, the paper lacks analysis of whether minority language subsets support reliable conclusions

- **Statistical Significance Testing**
The paper lacks statistical significance testing for reported performance differences, provides no confidence intervals, and offers insufficient error analysis across the diverse evaluation scenarios. With Hindi dominating 26.8% of QA pairs and other languages having minimal representation, the paper lacks analysis of whether minority language subsets support reliable conclusions

**Questions:**

- Why was **Surya OCR** excluded from the OCR evaluation? Given that Surya has demonstrated superior performance on Indic languages, its absence represents a significant gap in baseline coverage

- Authors mention adversarial questions "incorporate false assumptions". What methodology ensured these questions were appropriately challenging? Even Gemini-2.5 scores only 5.79/10 on adversarial questions - does this indicate poor question design or fundamental model limitations?

---

> ### Author Response · Authors · 2025-11-22
>
> We thank the reviewer for the thoughtful and constructive assessment of our work. We appreciate the recognition of the comprehensiveness and linguistic coverage of our work. We address the reviewer’s concerns in detail below.
>
> > **W1:** Limited Methodological Novelty
>
> We thank the reviewer for the feedback. While our benchmark leverages some established paradigms, its novelty lies in several key aspects:
>  - **Unified cross-modal, cross-lingual task suite**: VQA, MMT, and OCR tasks built around 5K+ unique images.
>  - **Cultural taxonomy**: 13 topics capturing region-specific practices.
>  - **Adversarial cultural questions**: Designed to probe failures and culturally grounded misconceptions.
>
> While LLMs were used to generate seed questions for efficiency, annotators were instructed to substantially revise or rewrite any culturally weak questions. Adversarial questions further test the model’s ability to reject false cultural assumptions.
>
> We also developed a custom pipeline for the OCR benchmark in Indic languages using human-verified ground truth from publicly available Wikisource documents. For this track, LLMs were not used explicitly at any stage to get the ground truth annotations.
>
> Finally, our controlled design enables systematic analysis of cross-lingual variation and region–language effects, offering insights into where multimodal LLMs succeed or fail across India’s diverse cultural and linguistic landscape. We also evaluate the necessity of visual grounding: Table 5 shows that removing images leads to a substantial drop in accuracy, highlighting the benchmark’s multimodal and culturally inclusive nature.
>
> > **W2:** Self-preferential bias & evaluation circularity
>
> We thank the reviewer for raising this concern. During data collection, annotators were explicitly instructed to fully revise or rewrite any questions lacking cultural grounding. Here, we also clarify the design safeguards we adopted to minimize the bias and the empirical evidence indicating that performance is not simply an artifact of bias:
>
> **1. Human rewriting substantially breaks the “generation–evaluation” loop**
>
> For transparency, Appendix E now reports edit rates and average relative edits, demonstrating substantial human intervention (18-79% across different tracks, more details in W4). These statistics indicate that the final VQA pairs are heavily refined beyond the raw LLM drafts, minimizing potential bias.
>
> **2. IVB-OCR contains zero LLM-generated content**
>
> A major component of the benchmark does not involve LLM seed generation at all. IVB-OCR is  based on Wikisource digitized scans where we find that Gemini-2.5 still achieves the strongest performance.
>
> **3. Different LLM-as-a-judge for evaluation**
>
> Additionally we deliberately ensured that the judge LLM used was different from the QA generation model family to avoid self-preferential scoring.
>
> Despite this, however, we also acknowledge that while LLM-generated seed questions help accelerate and guide the QA creation process, they may still introduce residual bias. We have included this discussion in the Limitations section (Appendix A) of the revised submission.

---

> > ### Author Response · Authors · 2025-11-22
> >
> > > **W3:** Absence of Design Choice Validation
> >
> > We thank the reviewer for raising this point. Following prior work [1], we used LLM-generated seed questions to accelerate the creation of 6 QA types per image, while explicitly instructing annotators to substantially revise or completely rewrite any questions that lacked cultural grounding. To validate our choice of the Gemini family for this initial synthesis step, we initially conducted a small scale human preference pilot study comparing Gemini-2.5-Flash and GPT-4o across English QA generation and Hindi/Telugu translations. Gemini showed comparable quality for English QA generations (48 vs. 34 votes) and clear advantages for translations (Hindi 40 vs. 25; and Telugu  55 vs. 23). Given this quality profile alongside significantly lower generation cost, we selected the Gemini family to generate seed QAs. We have added a brief discussion regarding this in the revised paper (Line 1399 in Appendix D).
> >
> > [1] All Languages Matter: Evaluating LMMs on Culturally Diverse 100 Languages Vayani et al. CVPR 2025
> >
> > > Linguistic distribution skew (e.g., Hindi at 26.8%)
> >
> > We want to emphasize that we used Indian states as a proxy for cultural groups. The distribution of languages in IndicVisionBench is a direct consequence of our state-wise, culture-first sampling strategy, where each state is mapped to its most widely spoken language. Because several Indian states (Uttar Pradesh, Madhya Pradesh, Rajasthan, Haryana, Bihar, Uttarakhand, and Delhi) are predominantly Hindi-speaking demographically, a higher proportion of Hindi QA pairs naturally emerges when aiming for balanced cultural coverage rather than language coverage. Thus, the images were annotated and translated according to the state’s primary language. Our goal was to reflect India’s cultural and geographic diversity rather than enforce an artificial language-uniform distribution. We have clarified this rationale and the design choice in the revised manuscript (Appendix D).
> >
> > > **W4:** Complete Lack of Quality Control Statistics:
> >
> > We thank the reviewer for highlighting the need to report quality control statistics. To address this, we calculated edit rates and average relative edits across all corpora. The edit rate measures the fraction of instances requiring human edits, while average relative edit quantifies the magnitude of changes. For each edited instance, we compute the Levenshtein distance between the edited and original text, normalize by the original text length, and average across all edited instances.
> >
> > Across the benchmark, human annotators made extensive revisions: English questions in VQA show an 18.61% edit rate (avg. relative edit 77.2%) while VQA-Indic shows 62.50% (48.9%). A relative edit value of 128% for Levenshtein distance indicates that the English Short-QA corrections were substantial, often exceeding the length of the original text and reflects major rewrites rather than minor adjustments. The MMT corpus also exhibits an edit rate of 79.1% and an average relative edit of 21.6% per caption, underscoring the human refinement involved in the translation process. These numbers indicate that human reviewers intervened heavily and that the final content is well refined beyond the raw LLM drafts. We have added these statistics, including language-level edits and annotator demographics in Appendix E of the revised manuscript for transparency.
> >
> > > **W5:** Inadequately Documented Subset Selection
> >
> > We thank the reviewer for raising this point. Our subset selection stems directly from the “state-wise” design of the benchmark, where each image is linked to a specific Indian region (state/UT) during collection. Because all the images were gathered with explicit state metadata wherever possible, we were able to map the majority of images to the state’s primary language, which then guided translation and annotation.
> > - VQA-Indic: The English questions were translated to create VQA-Indic, selected to ensure region-level balance. This ensured cultural and geographic diversity rather than random or convenience sampling. Many crowdsourced images already included cultural or regional attributes, making them natural candidates for Indic-language QA creation in the corresponding state language.
> > - VQA-Parallel (106 images): This subset was intentionally designed as a controlled multilingual evaluation set, not as a scale-oriented one. We selected 106 images such that (i) all 10 languages are covered with no explicit over-representation of a particular region and (ii) all 13 cultural topics were represented. This yielded a consistent, parallelizable set suitable for high-quality human translation into 11 Indic languages.
> >
> > We have clarified these selection criteria in the revised manuscript (in Appendix D) to make the subset construction fully transparent.

---

> > > ### Author Response · Authors · 2025-11-22
> > >
> > > > **W6:** Statistical Significance Testing
> > >
> > > We thank the reviewer for this important observation. We conduct Wilcoxon signed-rank tests for low-resource languages and also report 95% bootstrap confidence intervals for mean paired differences. Across all evaluated languages and question types, results are mostly statistically significant, with confidence intervals consistently above zero for both small (e.g., Punjabi) and large (e.g., Hindi) subsets. More details of these statistical analyses are included in Appendix C.1 (Tables 14, 15).
> > >
> > > In addition, to also evaluate the reliability of LLM-as-a-judge, we conduct a human study on two question types (Short-Answer and Adversarial) with two independent human annotators. We calculate the Pearson correlation where GPT-4o matches human–human agreement. For Short-Answer, human–human Pearson correlations range from r=0.66 to 0.68, while GPT-4o vs. human-mean correlations from r=0.74 to 0.79. For Adversarial questions, the range for human–human agreement is 0.64-0.74 while GPT-4o vs. human-mean is 0.73-0.80. These results indicate that GPT-4o achieves high correlation with human judges. We have added the experimental setup and discussion in Appendix C.2.
> > >
> > > > **Q1:** Why was Surya OCR excluded?
> > >
> > > We appreciate the reviewer bringing this to our attention. Indeed, Surya OCR is a strong baseline that has now been added to our evaluation, and its results are reported in Table 4 of the revised version to provide a complete and fair comparison of different OCR baselines. Surya ranks second best to Gemini-2.5 in Gujarati, Hindi, Kannada and Marathi. We thank the reviewer for making the choice of models more comprehensive and our work stronger.
> > >
> > > > **Q2:** Adversarial questions
> > >
> > > We thank the reviewer for this thoughtful question. Adversarial questions were constructed using a two-stage methodology: (i) seeding culturally grounded false premises via an LLM (e.g., incorrect festival associations or misattributed regional practices), and (ii) requiring annotators to explicitly identify, reject, or correct the false assumption rather than simply answer. An example from our dataset: “What kind of rituals are performed for the Hindu deity Krishna in this temple?” where the image depicts a Jain temple, not a Hindu temple. All adversarial prompts were thus manually validated to ensure the false presuppositions are realistic, culturally plausible, and not trivially detectable. The relatively low scores are thus expected and reflect model limitations. These questions specifically test challenging multimodal reasoning skills: detecting presupposition errors, handling culturally specific counterfactuals, and rejecting misleading cues. We have clarified this methodology and added a discussion (Appendix D) explaining why adversarial performance highlights genuine gaps in current models.

---

> ### Author Response · Authors · 2025-11-27
>
> Dear reviewer,
>
> Thank you for your detailed and constructive review. Your feedback significantly strengthened our work, particularly through the inclusion of:
> - additional baselines (Surya OCR),
> - quality-control statistics like edit rate and average relative edit,
> - statistical significance tests and confidence intervals,
> - further clarifications regarding benchmark construction
>
> We would like to confirm that these revisions satisfactorily address all of your concerns.

---

### Official Review · Reviewer_f28F · 2025-11-01

**Soundness:** 3
**Presentation:** 3
**Contribution:** 3
**Rating:** 8
**Confidence:** 4

**Summary:**

This paper proposes a human-annotated multimodal benchmark for Indic languages. The benchmark includes tasks such as OCR, MT, and visual QA. The authors evaluated several vision-language models, including both open-source and commercial ones, and identified a notable performance gap.

**Strengths:**

This work addresses an underrepresented language and provides a valuable resource for the multilingual AI community, particularly for the Indic community. In the current era of LLM-generated and synthetic data, I appreciate the thorough effort to involve humans in data creation. The resulting dataset covers multiple tasks and languages. The annotation guidelines and interface are also clearly described, further demonstrating the authors’ commitment to transparency.

**Weaknesses:**

Annotator demography is missing, which will be useful information to add in this type of dataset.

**Questions:**

-

---

> ### Author Response · Authors · 2025-11-21
>
> We sincerely thank the reviewer for their positive and encouraging assessment of our work. We appreciate the acknowledgement of the need for Indic-language benchmarks, the emphasis placed on the value of human involvement in our data creation pipeline, and the recognition of our commitment to transparency. We are grateful that the reviewer views IndicVisionBench as a meaningful contribution to the multilingual AI community.
>
> > **W1:** Annotator demography is missing
>
> We thank the reviewer for highlighting the importance of reporting annotator demographics, especially for culturally grounded datasets. We fully agree that such information enhances transparency and contextualizes the dataset’s cultural coverage. Below, we provide the demographic details for both groups involved in data creation: (i) crowd-sourced image contributors and (ii) the QA correction/translation team.
>
> *Crowdsourced Image Contributors (n = 18)*
>   - Gender: 13 male, 5 female
>   - Age distribution: 20–30 (10), 30–40 (6), 40–50 (2)
>
> *QA Correction / Translation Team (n = 25)*
>   - Gender: 13 female, 12 male
>   - Age distribution: 20–30 (17), 30–40 (8)
>   - Language distribution: 8 for English, 3 for Hindi, and 1–2 annotators for each of the remaining Indic languages in the benchmark.
>
> We have also added the corresponding details in Appendix E (Figure 23), summarizing these demographic distributions in the revised submission.

---

> > ### Author Response · Authors · 2025-11-27
> >
> > Dear reviewer,
> >
> > Thank you again for your positive and encouraging review. We are grateful for your time and insights for improving this work. We hope that all your comments have been fully addressed.

---

### Official Review · Reviewer_HKVC · 2025-11-01

**Soundness:** 3
**Presentation:** 3
**Contribution:** 3
**Rating:** 6
**Confidence:** 4

**Summary:**

This paper addresses the Western-centric bias in existing vision-language model evaluation benchmarks by introducing IndicVisionBench. This new benchmark is the first of its kind to focus on the cultural and linguistic diversity of the Indian subcontinent, covering English and 10 Indic languages. The benchmark is composed of 5K images and over 37K question-answer pairs, structured into three distinct tasks: VQA, OCR, and multimodal MT. The data curation process involved a combination of web crawling, crowdsourcing, and rigorous human verification to ensure cultural and linguistic fidelity. The authors evaluate eight prominent VLMs, from proprietary systems to open-weight models, and reveal significant performance deficits, particularly for low-resource languages and culturally nuanced queries. The analysis delves into regional biases, cross-lingual performance variations, and topic-specific capabilities, demonstrating the benchmark's utility in uncovering the limitations of current state-of-the-art multimodal systems.

**Strengths:**

- The authors' motivation for this work isintuitive They address a well-known but often-neglected limitation in our field: the Western-centric nature of most vision-language evaluation benchmarks. The paper makes a very convincing argument for the necessity of developing resources that capture greater cultural and linguistic diversity, a direction of research that is becoming increasingly critical.

- A major strength of this paper is the comprehensive design of IndicVisionBench. It is not limited to a single task but integrates three distinct multimodal evaluations (VQA, MMT, and OCR) especially the MT part, which makes it a very comprehensive resource. The VQA component is especially well-designed, featuring six different question formats. The use of "adversarial questions" to probe for the ability to reject false cultural assumptions is a particularly novel and insightful method for assessing model robustness.

- The experimental evaluation presented is comprehensive, covering a wide range of current vision-language models. Importantly, the analysis goes beyond a simple ranking of models. The authors provide valuable insights through detailed breakdowns of performance, for example by investigating regional-language biases and performance variations across different cultural topics.

**Weaknesses:**

- There is a potential for bias in the data generation process. The VQA pairs were first generated using Gemini models and then corrected by human annotators. While this is a practical and common approach, it introduces a risk of "self-enhancement" bias, where the models used for generation might produce content that is easier for them to evaluate later. The paper would be strengthened by acknowledging and discussing this potential limitation.

- The evaluation of open-ended questions relies on GPT-4o as a judge. Such judges can have their own biases and may not fully capture all cultural nuances, introducing a layer of uncertainty. It would be beneficial to include a brief discussion on these known limitations. Reporting inter-annotator agreement from a small subset of human judges, to calibrate the LLM judge's performance, would also add significant value.

- The benchmark is presented as "large-scale," but the amount of data for some specific languages and tasks is quite small. For instance, the VQA-Parallel and MMT tracks are built upon only 106 images. For some of the lowest-resource Indic languages, the number of samples is also limited. This can affect the statistical significance of the conclusions for these specific languages. The authors should be cautious with making very strong claims about performance on languages with a small number of samples.

- A very interesting point is made in Section 6 regarding the limitations of WER/CER metrics for OCR evaluation, and the authors make a good case for using ANLS instead. This discussion is valuable. However, it appears quite late in the paper. The impact would be greater if this challenge were introduced earlier, in the experimental setup (Section 4), to provide better context for the results presented in Table

**Questions:**

No

---

> ### Author Response · Authors · 2025-11-22
>
> We sincerely thank the reviewer for the thoughtful and constructive feedback. We are grateful for the recognition of our motivation, the comprehensive design, cultural and linguistic breadth of IndicVisionBench. We also appreciate the reviewer highlighting the novelty and the overall value of our benchmark to the community.
>
> Below, we address each of the reviewer’s concerns and outline clarifications and updates we included in the revised version.
>
> > **W1**: Potential bias due to Gemini-generated VQA items and “self-enhancement” effects
>
> We thank the reviewer for raising this concern. In our data collection, annotators were explicitly instructed to fully revise or rewrite any questions lacking cultural grounding. However, we also acknowledge that while LLM-generated seed questions help accelerate QA creation, they may still introduce residual bias. As suggested, we have included this discussion in the Limitations section (Appendix A) of the revised submission. For transparency, Appendix E now reports edit rates and average relative edits, demonstrating substantial human intervention. These statistics indicate that the final VQA pairs are heavily refined beyond the raw LLM drafts, minimizing potential bias.
>
> > **W2**: Reliability and bias of GPT-4o as an open-ended judge
>
> We thank the reviewer for raising this important concern. We use GPT-4o as an LLM-as-a-judge to remain consistent with prior work [1] and to ensure independence from the Gemini family, which generated the initial QA drafts. As suggested, we conducted a small human evaluation on two question types Short-Answer and Adversarial on the English VQA-Parallel subset with two annotators.
>
> GPT-4o matches human–human agreement. For the Short-Answer question, human–human Pearson correlations range from r=0.66 to 0.68, while GPT-4o vs. human-mean correlations from r=0.74 to 0.79. For Adversarial questions, the range for human–human agreement is 0.64-0.74 while GPT-4o vs. human-mean is 0.73-0.80. These results indicate that GPT-4o achieves high correlation with human judges. We, however, also acknowledge that LLM judges can have biases and may not fully capture all cultural nuances with the experimental setup and discussion added in Appendix C.2.
>
> [1] All Languages Matter: Evaluating LMMs on Culturally Diverse 100 Languages Vayani et al. CVPR 2025
>
> > **W3**: Small sample sizes in some languages and tasks
>
> We thank the reviewer for this important observation. We have updated the manuscript to exercise caution in making strong claims for languages and tasks with limited samples. We further conduct Wilcoxon signed-rank tests for low-resource languages and also report 95% bootstrap confidence intervals for mean paired differences. Across all evaluated languages and question types, results are mostly statistically significant, with confidence intervals consistently above zero for both small (e.g., Punjabi) and large (e.g., Hindi) subsets. More details of these statistical analyses are included in Appendix C.1 (Tables 14 and 15).
>
> > **W4**: ANLS related discussion
>
> We thank the reviewer for this helpful suggestion. As recommended, we have added an initial discussion of ANLS and its motivation in Section 4 (Line 270) to provide clearer context before presenting the OCR results. We thank the reviewer for making our work stronger and are encouraged by overall positive feedback.

---

> > ### Author Response · Authors · 2025-11-27
> >
> > Dear reviewer,
> >
> > Thank you again for your thoughtful and constructive review. We appreciate your positive assessment of our motivation, benchmark design, overall contribution and we are grateful for your time and insights. We hope that all your concerns have been fully addressed in our rebuttal and revisions.

---

### Author Response · Authors · 2025-12-03
**Rebuttal Summary**

We sincerely thank the reviewers for their thoughtful and constructive feedback. Their comments have substantially improved the clarity, rigour, and transparency of the paper. Below, we first summarize the reviewer ratings:
- **Reviewer f28F - Rating: 8, Confidence: 4**
- **Reviewer w7Rf - Rating: 8, Confidence: 4**
- **Reviewer HKVC - Rating: 6, Confidence: 4**
- **Reviewer iVmL - Rating: 2, Confidence: 4**

Three reviewers (each with confidence 4) provided positive scores of **8, 8, and 6**, indicating strong alignment regarding the contribution and quality of the work.

**Key strengths recognized by reviewers**
- **Clear Motivation and Cultural Importance:** Reviewers highlighted the significance of addressing the lack of culturally contextualized multimodal benchmarks and appreciated the coverage across 10 Indic languages and 13 cultural topics.
- **Comprehensive and Well-Structured Benchmark:** The multi-task design (VQA, MMT, OCR), inclusion of six VQA questions and the human involvement in data refinement were explicitly praised. Reviewers also commended the construction of the VQA-Parallel and MMT subsets for analyzing cross-lingual effects.
- **Insightful and Rigorous Evaluation:** Reviewers noted that the evaluation across eight VLMs, along with analyses of regional–language biases, cultural topic differences, and robustness to false assumptions, provides meaningful insights that go beyond model ranking.

**Response for main concerns**
- **Quality-control statistics and Edit rates:** We report edit rates and average relative edits across all corpora, with detailed breakdowns across QA types and languages (Appendix E). We clarify the extent of human rewriting in the updated submission.
- **Statistical significance and confidence intervals:** We include Wilcoxon signed-rank tests and 95% bootstrap confidence intervals across both low-resource languages and high-resource languages (Appendix C.1, Table 14, Table 15).
- **Reliability of LLM-as-a-judge (GPT-4o):** We also conduct a two-annotator human evaluation on different questions in VQA-Parallel. GPT-4o achieves human-level consistency and correlates highly with human judgement (Appendix C.2).
- **Validation of model choice for seed QA generation:** We add results of a pilot human preference study comparing different models for English QA generation and Hindi/Telugu translations. Gemini models demonstrate comparable or superior quality while offering lower cost (Appendix D).
- **Benchmark construction details and annotator demographics:** We expand our discussion of adversarial question construction, state-wise sampling strategy, and the cultural–linguistic mapping that explains the proportion of Hindi examples. We also document annotator demographics for image contributors and QA correction teams (Appendix E, Figure 23).
- **Evaluating potential pretraining overlap:** We assess whether the strong performance of certain models could be influenced by pretraining overlap. The IVB-OCR subset uses Wikisource page scans, and the IVB-VQA dataset includes 615 crowdsourced images that are not indexed online. Model performance trends remain consistent on these demonstrably unseen subsets (Tables 4 and 6).
- **Additional baselines:** We additionally incorporate results from the Surya model in the OCR evaluation for completeness (Table 4).
---
We thank all reviewers again for the time, care, and insight they have invested in strengthening this work. We hope that the revisions and additional analyses fully address the raised concerns.

---

### Meta-Review · Area_Chair_VjSm · 2026-01-08

**Summary:**

**Strengths**:

1. Addresses an important gap by providing a culturally grounded, multilingual, and Indic-centric benchmark, counterbalancing Western-centric evaluation settings.

2. Covers three multimodal tasks (VQA, OCR, MMT) and 10 Indic languages, enabling broader analysis than single-task or single-language datasets.

3. Includes adversarial question types, cultural topics, and a parallel multilingual slice, enabling controlled cross-lingual comparisons and robustness probing.

4. Evaluation includes both closed and open models, providing informative comparisons for the community.

5. Annotation pipeline involved human refinement and guidelines, improving cultural and factual accuracy relative to purely synthetic data.

**Weaknesses**:

1. Heavy dependence on Gemini models for synthetic generation, raising concerns about evaluation circularity, bias inheritance, and “self-preferential” performance.

2. Scale and coverage limitations: some languages and tasks have very small sample counts (e.g., 106-image subsets), limiting statistical robustness.

3. Lack of quality control reporting, including missing statistics on human edit rates, inter-annotator agreement, and cultural alignment adjustments.

4. Reliance on GPT-4o as an evaluator introduces judging bias and cultural nuance limitations; no cross-judge or human calibration analysis is provided.

5. Benchmark construction methodology follows existing paradigms and is incremental in novelty, focusing more on expansion of scope than methodological innovation.

6. Demographic and dataset sourcing transparency is limited, including annotator demographics and Google-search image provenance concerns.

**Reviewer Concerns:**

Most concerns have been addressed.

**Not fully resolved:**

The benchmark construction largely follows existing paradigms and is incremental in novelty, expanding scope rather than introducing methodological innovations.

**Reviewer Scores:**

- Reviewer HKVC: 6 -> 6
- Reviewer f28F: 8 -> 8
- Reviewer iVmL: 2 -> 2
- Reviewer w7Rf: 8 -> 8

---

### Decision · Program_Chairs · 2026-01-26

Accept (Poster)